

# Research trends on hazards, disasters, risk reduction and climate change in Indonesia: a systematic literature review

Riyanti Djalante[1,2]

[1]Alexander von Humboldt Experienced Researcher Fellow / Research Associate, UNU-EHS, UN Campus, Platz der Vereinten Nationen 1, Bonn 53117, Germany
[2] Honorary Lecturer, University of Halu Oleo, Sulawesi Tenggara, Indonesia

*Correspondence to:* Riyanti Djalante (djalante@ehs.unu.edu)

**Abstract**. The number of disasters due to natural hazards and climate change are on the rise. Within the last decade the world has experienced the most frequent and impactful disasters. The year 2015 was the hottest year ever and the associated
disaster impacts have drastically increased the cost to the society socially and economically. The Asia Pacific region has been the place where these disasters occur the most. Indonesia, one of the countries in this region, is one of the most at risks from disasters and climate change impacts in this region.

This paper aims to do a systematic literature review on published academic materials related to hazards, risks, disaster risks reduction (DRR) and climate change in Indonesia. Systematic literature review is defined as systematic or evidence-based
literature reviews with explicit and transparent methods and follows a standard protocol or a series of stages so that bias can be reduced and more importantly able to provide a comprehensive body of knowledge. While there is a vast material that have been published related to hazards and DRR on Indonesia, there has not yet a literature review that examines these materials in a comprehensive and systematic way. This systematic review is important since it outlines recent research progress over time which can help to determine which topics have been heavily researched and thus seeks to recommend
future research needs. The author conducts a multi-staged literature review to study publications that are indexed within SCOPUS. Multi-stage processes are taken to determine inclusion and exclusion for more relevant findings. The author also consults authors' and organizations profiles from Google Scholar, Research Gate, to determine gender, affiliations, and extent of publications.

The first stage of search from Scopus gives a list of 5253 publications by which after second stage gives 1478 publications
and third stage gives a final most relevant publication of 744. The findings are outlined in two parts. One on the results of the analysis in terms of times of publications, most active researchers and research organizations, most cited papers, and categorization of major research topics. The other one is on the examinations on the roles of Indonesian authors and organizations in publishing in international journals, involvement in highly cited papers, and how collaborations have taken place amongst Indonesian and international researchers and organizations. This thus led to recommendations for capacity
building in research in Indonesia.

The findings on the first part are as follow. The final selected publications are categorized into three major topics of (1) hazard, risks and disaster assessments (HRD), (2) disaster risk reduction (DRR), and (3) climate change vulnerability, impacts and adaptation (CC). Publications on the category of HRD are comprised of more than half of the total publications, while the rest is divided amongst those related to DRR and CC. The oldest publication was issued in 1978 and the earlier
period publications were heavily focused on the topics of geophysical hazards and risks related to earthquake, volcanic activity and tsunami. There were a surge of publications following the 2004 Indian Ocean tsunami which impacted Aceh while publications related to DRR and CC increasingly gaining ground in the last 10 years. A more detailed analysis on research topics shows that on the HRD group is mainly related to research on volcanic eruption, tsunami and earthquake. Research on the DRR group focuses on governance, recovery and reconstruction, early warning systems. Those on CC
groups, the research are mainly on reducing emissions from deforestation and forest degradation, governance of adaptation and climate change impacts on different sectors.





The findings on the role of Indonesian researchers and research organizations show great needs for capacity building in research, publications and collaborations. The study finds that international non-Indonesia authors dominate the number of researchers. Only half of the publications are co-authored by Indonesians. Collaborations have indeed taken place amongst

between international and Indonesian organizations but it is only by limited number of Indonesian organizations or researchers. This suggest that Indonesians researchers tend to work with other Indonesians and hence needed to expand their collaborations with international scholars as a strategy to increase the quality of the publications measured by the number of citations and ability to submit for higher impact journals.

The paper recommends further research to be done on research on hazards and risks identifications on other locations in

Indonesia, preparedness and on vulnerable groups, and governance and impacts of climate change on different sectors. It also calls for more strengthening capacity of Indonesian authors in writing for international journal publications and creating space for collaborations amongst Indonesian and international researchers.

**Acknowledgement**. The author would like to express its gratitude to the Alexander von Humboldt Foundation through its

Fellowships for Experienced Researcher, which has enabled the author to conduct a research visit for 18 months in Germany. The author would also like to thanks Dr Matthias Garschagen for his earlier review of the manuscript. This paper reflects to author own view and not representing any organization.

**Keywords**. Systematic literature review; Indonesia; disaster; natural hazard; climate change

**1    Introduction**

Disaster events and their associated social and economical impacts are on the rise. The last decade has shown the highest number and impacts from disasters while 2015 has been stated as the hottest year ever. The Asia Pacific region has been the place where these disasters occur most while Indonesia is one of the most at risks from disasters and climate change impacts (Figure 1).

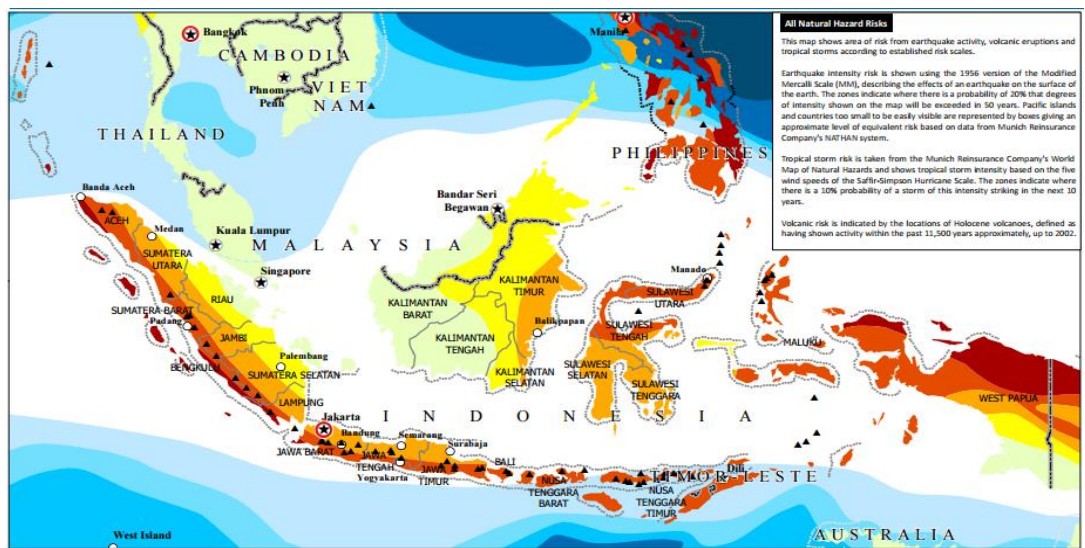


**Figure 1 Hazard map of Indonesia (OCHA-ROAP 2011)**

Over the last century, there have been 429 disasters caused by natural hazards, more than 200 thousands death, more than 29 million people in total affected and the total damage is above 44million USD (Table 1) (EMDAT, 2016).



**Table 1 Disaster impacts in Indonesia from 1900 - 2016 (EMDAT, 2016)**

| Disaster type | Occurrence | Total deaths | Affected | Injured | Homeless | Total affected | Total damage (USD) |
|---|---|---|---|---|---|---|---|
| Earthquake | 115 | 198487 | 7401192 | 171429 | 1556548 | 9129169 | 11695926 |
| Volcanic activity | 56 | 18310 | 1294297 | 3731 | 23500 | 1321528 | 530390 |
| Drought | 10 | 9340 | 4804220 | 0 | 0 | 4804220 | 160200 |
| Flood | 172 | 6555 | 9445598 | 255197 | 183295 | 9884090 | 6422047 |
| Landslide | 53 | 2423 | 356696 | 540 | 40015 | 397251 | 120745 |
| Mass movement (dry) | 1 | 131 | 651 | 50 | 0 | 701 | 1000 |
| Storm | 12 | 2013 | 28715 | 243 | 1290 | 30248 | 1000 |
| Wildfire | 10 | 319 | 3443664 | 478 | 0 | 3444142 | 25429000 |
| Total | 429 | 237,578 | 26,775,033 | 431,668 | 1,804,648 | 29,011,349 | 44,360,308 |

Furthermore, when comparing the impacts between geophysical and those hydro-meteor-climato-logical disasters, while disasters caused by climate occurs and impacts more, the number of deaths is significantly caused by earthquake and volcanic activities (Figure 2). Hence, it is important to differentiate the hazard types but also to integrated risks management

from both types in an integrated fashion (e.g. Djalante and Thomalla, 2012; Thomalla et al., 2006).

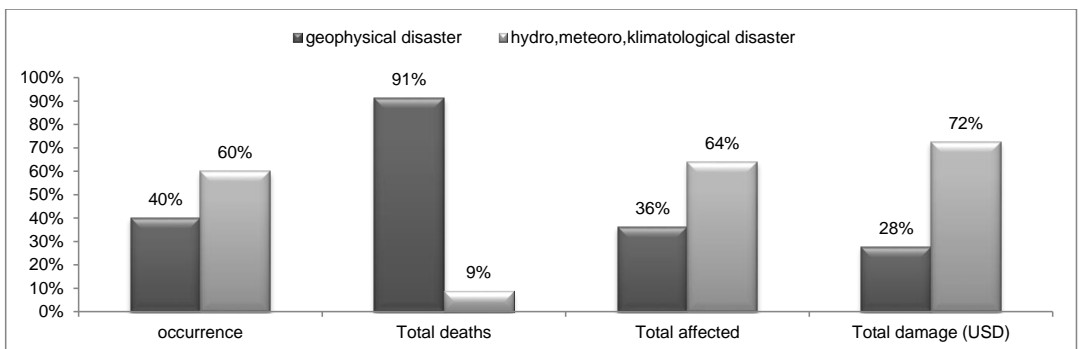

**Figure 2 Comparing between the impacts of geophysical and hydro-meteoro-klimatological disasters (modified from EMDAT, 2016)**

This paper aims to systematically review literature on related to hazards, risks and disaster risks reduction, and climate change vulnerability, impact, and assessments in Indonesia. Systematic literature review is briefly defined as a method to systematically reviewing evidence or literature with explicit and transparent methods. A systematic review method has been used widely in the field of health (Moher *et al.*, 2009), software engineering (Kitchenham *et al.*, 2009), and engineering (Carrion and Levinson, 2012; Chai et al., 2013; Gosling and Naim, 2009; Liang et al., 2009). Despite the importance of

systematic literature review, there have been few studies that use this in the topic related to hazards, disasters, and or climate change. Some notable examples are review on drought (Woodhouse and Overpeck, 1998), landslide (Aleotti and Chowdhury, 1999), contaminants (Noyes *et al.*, 2009), urban ecosystem (Luederitz *et al.*, 2015), ecosystem-based adaptation (Brink et al., 2016; Kabisch et al., 2015). A notable study on systematic review of climate change is done by Berrang-Ford et al (Berrang-Ford et al., 2014; Berrang-Ford et al., 2011; Berrang-Ford et al., 2015; Ford et al., 2015a; Ford et al., 2015b;

Ford et al., 2013; Ford et al., 2011; Ford et al., 2012; Lesnikowski et al., 2013a; Lesnikowski et al., 2013b; McLeman et al., 2014; Paterson et al., 2012; Pearce et al., 2011; Thompson et al., 2010).



Even though there is a vast material that have been published related to this topic on Indonesia, there has not yet a literature review that examines these materials in a comprehensive and systematic way. By reviewing published works in this fashion, researchers can build upon others´ works and avoid reinventing the wheel so that not only determining which areas and topics that have been heavily researched, but also which further areas that needed more researches. There are two research questions adopted. **First** is on progress of research on hazards, risks, disasters and climate change in Indonesia, and when, how and who have been involved in those research and publications. **Second** is on roles of Indonesian authors in contributing for research, publishing in international journals, involvement in highly cited papers, and collaborations amongst Indonesian and international researchers and organizations. The key argument of this paper is that while there are limited number of Indonesian authors who and research organizations that have collaborated and published globally, in general, Indonesian authors have lower level of involvement in international collaboration and publications in high quality and high impact publications. Based on their extensive review on climate change literature, Berrang-Ford et al (2011; 2015) suggested an analytical approach for systematic review and research synthesis as presented in Table 2, which is adopted in this paper.

**Table 2 Analytical approach for the systematic review**

| Topics | Descriptions |
|---|---|
| **Research questions and aim** | · Explicit<br>· Clear description |
| **Data sources and document selection** | · Justification and description of sources<br>· Articulation of search term<br>· Description of inclusion and exclusion<br>· Documentation of literature included and excluded |
| **Analysis and presentation of results** | · Description of method for analysis<br>· Critical appraisal of information quality |

The structured of the paper follows the above analytical approach. The first section of this paper outlines the rationale, aim and research questions adopted. The second section outlines research method related to data sources and document selection. The third section gives the analysis and presentation of results, and the last section describes the conclusion and recommendations for further research.

## 2    Research method: data sources and document selection

### 2.1    Justification and description of sources

The author conducts a multi-layered literature review to study publications using the Scopus research engine. There have been several studies comparing strengths and weakness of PubMed, Scopus, Web of Science and Google Scholar (Bakkalbasi et al., 2006; Bar-Ilan, 2008; Falagas et al., 2008; Kulkarni et al., 2009). Scopus research engine is selected because it is the largest abstract and database of peer-review literature (Burnham, 2006; De Moya-Anegón et al., 2007; Leydesdorff et al., 2010). Additional information is gathered from Google Scholar (Google, 2016), Research Gate (Gate, 2016) or researchers´ profiles to give the full extent of particular scholars' works. The author checks the organizations, nationalities and genders of the researchers in the Internet through Google. Multi-staged processes are taken to determine inclusion and exclusion for more relevant findings.

### 2.2    Articulation of search term and description and documentation of inclusion and exclusion

**First stage**



The author input the following search terms into SCOPUS which gives a total hit of 5253 publications, (TITLE-ABS-KEY(hazard*) OR TITLE-ABS-KEY(risk*) OR TITLE-ABS-KEY(disaster*) OR TITLE-ABS-KEY(disaster management*) OR TITLE-ABS-KEY(disaster risk reduction*) OR TITLE-ABS-KEY(climate change*) OR TITLE-ABS-KEY(climate change adaptation*) OR TITLE-ABS-KEY(resilien*) AND TITLE-ABS-KEY(Indonesia)).

**Second stage**

The author applies the second stage to further refine the results. This gives a total hit of 1748 publications. The exclusion includes refinement in subject areas, in document types, in language (only in English and Bahasa Indonesia), and source title that does not directly related to the topic in DRR in Indonesia.

**Third stage**

The third layer search involve the author download the results into xml format, save it and import it into Microsoft Excel,

with using all delimiters factors. The results in the Excel format are examined line by line to further determine exclusion from the lists. Materials that are excluded in this final round is related to analysis of research in mining industry in Indonesia, those that discuss on the science of climate change and those that touch on the issue on disasters but not directly on Indonesia and when the author judges that the scope is too broad to be included are finally 744 materials selected. The final list is analyzed in terms of authorships, references, citations, keywords, places of focus, types of publications, impact factors,

time of publications and topics and sub-topics of research. Table 3 shows the EMDAT-CRED categorization of disaster groups and hazards that are used in this study to help more details analysis related to major research topics. Natural disaster groups caused by geophysical, meteorological, hydrological, and climatologically hazards are included. Those excluded are disasters caused by biological, extra-terrestrial and technological hazard.

**Table 3 Categorization of disaster groups included in this study (Source: EMDAT-CRED, 2016)**

| Disaster Group | Disaster Subgroup | Definition | Disaster Main Type | Disaster Sub-Sub-Type |
|---|---|---|---|---|
| **Natural** | Geophysical | A hazard originating from solid earth. This term is used interchangeably with the term geological hazard. | Earthquake | Ground shaking, tsunami |
| | | | Mass Movement | |
| | | | Volcanic activity | Ash fall, lahar, Pyroclastic flow, Lava flow |
| | Meteorological | A hazard caused by short-lived, micro- to meso-scale extreme weather and atmospheric conditions that last from minutes to days. | Extreme Temperature | Cold wave, heat wave, severe winter conditions |
| | | | Fog | |
| | | | Storm | Extra-tropical storm, Tropical storm, Convective Storm (Derecho, Hail, Lightning/thunderstorm, Rain, Tornado, Sand/dust storm, Winter storm/blizzard, Storm/surge, Wind) |
| | Hydrological | A hazard caused by the occurrence, movement, and distribution of surface and subsurface freshwater and saltwater. | Flood | Coastal flood Riverine flood Flash flood Ice jam flood |
| | | | Landslide | Avalanche (snow, debris, mudflow, rockfall) |
| | | | Wave action | Rogue wave, seiche |
| | Climatological | A hazard caused by long-lived, meso- to macro-scale atmospheric processes ranging from intra-seasonal to multi-decadal climate variability. | Drought | |
| | | | Glacial Lake Outburst | |
| | | | Wildfire | |





### 2.3 Critical appraisal of information quality

After the second stage is done, the author downloads to material into xml format and later imports it into the Microsoft Excel format. When importing into the excel format the author choose all delimiters to enable particular information goes to the right column. However, the results are not always consistent and hence a manual check on each entry row needed to be done.

Data from Scopus is used to determine which of the publication is highly cited, who are the most active authors and organizations, where they are based and what keywords are used. However the author finds that the number counts on the authors´ publications and citations presented in the SCOPUS search is sometimes different to the actual check of the excel sheet. It is also different when examining the profile of one particular author. Hence, to ensure consistency, the number of counts obtained from the list in excel sheet is used.

Moreover, the author crosschecks the number of citations from Scopus to the Internet, and adopts the higher citation counts. It is generally the case that data from Google search on the publication and author leads to higher and more up to date citations counts. The author also consult total citations and publications of researchers in Google Scholar or Research Gate or Researcher other profile to make sure that the full list of publications are captured.

### 3 Analysis and Presentation of Results

This section is structured based on the research questions on the analysis of the materials and second on the roles of Indonesian authors and organizations. Materials gathered are analyzed in terms of time, active researchers, research organizations, most cited papers, and categorized into several main research topics. In each of main research topics, the analysis differentiates the roles of international and Indonesian researchers and research organizations.

### 3.1 Review of key research topics

#### 3.1.1 Groupings of topics

The author categorizes the final list into three groups in order to show and outline how changes in directions on research have taken place over the years and to reduce heavy unbalance towards findings on hazard and risks assessments toward earthquake and volcanic eruption research (Table 4).

**Table 4 Classifications of findings based on topics of research**

| Major topics groups | Relevant Definitions (UNISDR, 2009) |
|---|---|
| (1) hazard, risks, disasters assessments (HRD) | -Hazards: A dangerous phenomenon, substance, human activity or condition that may cause loss of life, injury or other health impacts, property damage, loss of livelihoods and services, social and economic disruption, or environmental damage.<br>-Risks: The combination of the probability of an event and its negative consequences.<br>-Disaster: A serious disruption of the functioning of a community or a society involving widespread human, material, economic or environmental losses and impacts, which exceeds the ability of the affected community or society to cope using its own resources. |
| (2) disaster risk management or reduction (DRR) | - The systematic process of using administrative directives, organizations, and operational skills and capacities to implement strategies, policies and improved coping capacities in order to lessen the adverse impacts of hazards and the possibility of disaster (UNISDR).<br>- The concept and practice of reducing disaster risks through systematic efforts to analyze and manage the causal factors of disasters, including through reduced exposure to hazards, lessened vulnerability of people and property, wise management of land and the environment, and improved preparedness for adverse events. |
| (3) climate change vulnerability, impacts and adaptation (CC) | -A change of climate which is attributed directly or indirectly to human activity that alters the composition of the global atmosphere and which is in addition to natural climate variability observed over comparable time periods (UNFCCC).<br>- The adjustment in natural or human systems in response to actual or expected climatic stimuli or their effects, which moderates harm or exploits beneficial opportunities (UNISDR). |



### 3.1.2    Yearly assessments

There are several periods of development in the publications, which are thought to be corresponded to the occurrence on major hazards or disasters events in Indonesia (Figure 4).

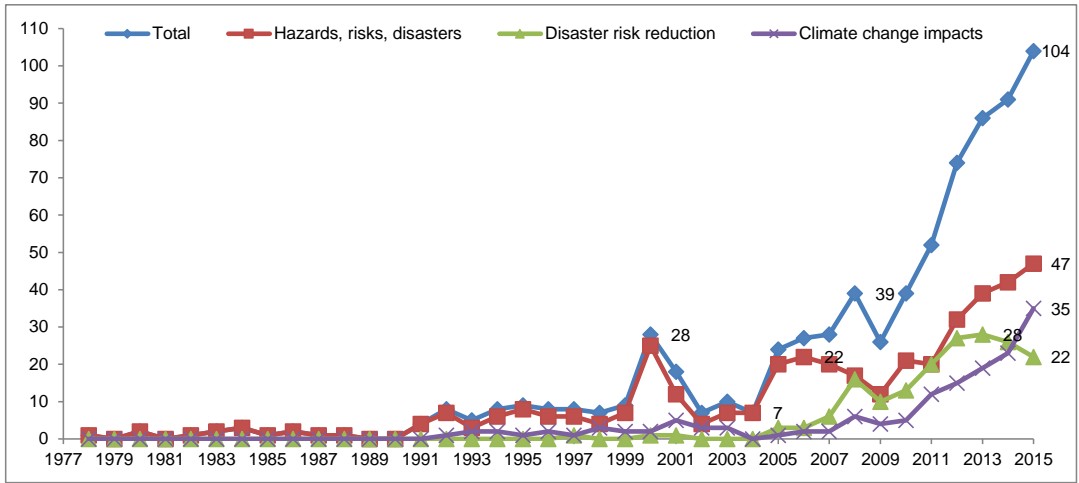


**Figure 3 Number of publications over the year (total 744)**

The first period is within the 1970s-1980s period. In this period, there were no significant changes in the numbers of publications produced. Researches in this period were heavily done on the topics of geophysical hazards and risks related to earthquake and volcanic eruption. The second period 1990s to 2000s shows a notable increase in literature where on average there were 10 publications per year. This gradual increase in literature mainly corresponds to the literature related to the assessments of hazards, risks and disasters and there is a sharp increase in literature which reached its highest point in 2000. The third period of 2000-2010s was the most dynamic period within the publications on literature. While there was a sharp decline since it reached its first peak in 2000, a surge of publications was started in 2004 which correspond to the Indian Ocean tsunami which hit Indonesia the most. This increase continues ever since. This is also the period when not only publications related to understanding the risks of earthquake and tsunami, but also those related to examining DRR and climate change impacts. The peak occurs between 2010 and 2015 which shows soaring published materials in all topics. There are 104 publications in 2015 which is the highest ever produced in a single year. In this period, publications related to climate change and their impact on Indonesia has started to be considered and is expected to still increase in the future. While both publications on hazards group and climate change group are expected to rise, the publications on the DRR shows a trend of decline.

### 3.1.3    Number of publications, citations, keywords and locations of research

As can be seen from Table 5, publications in the research topic related to hazards, risks, and disasters outweigh the other two categories. There are more than half materials are written on the topic of hazards, risks and disasters, and the rest is divided almost equally between those on DRR and climate change. The hazards, risks and disasters category also have the highest total numbers of citations, and have more than two third of the citations. An examination on the citation average however show a quite different story, while the climate change literature category has the least number of papers published, the citation average reach 8.0, which is similar to that of the hazard, risk and disaster category (Figure 4).





**Table 5 Total numbers of papers, citations and citation average**

| Main research topics | Numbers of papers | Numbers of citations | Citation average |
|---|---|---|---|
| **Hazards, risks, disasters (HRD)** | 412 | 3386 | 8.0 |
| **Disaster risk reduction (DRR)** | 177 | 668 | 3.8 |
| **Climate change (CC)** | 154 | 1237 | 8.0 |
| **Total** | **744** | **5291** | **-** |

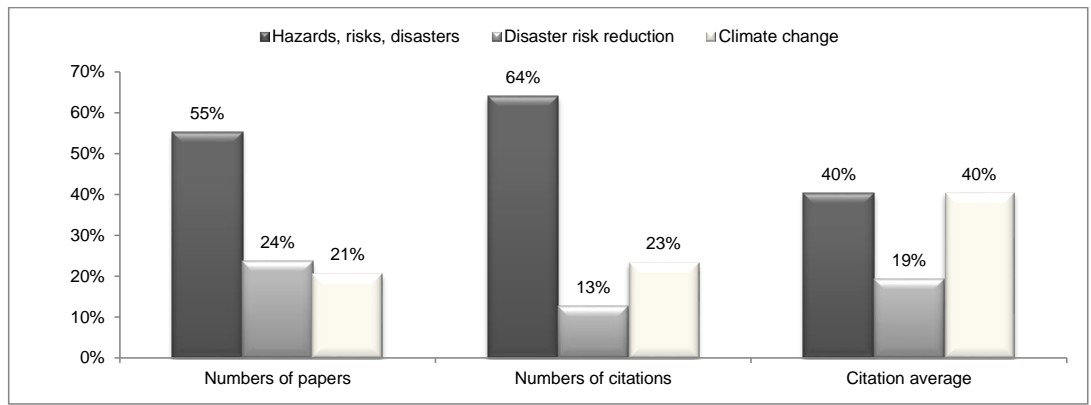

**Figure 4 Comparing publications in 3 categories in terms of numbers of papers, numbers of citations, and citation average per paper (total 744)**


A more detailed examinations on the keywords used are mostly related to place followed by those related to hazards, and risks and disasters (Figure 5). If we look at the locations within Indonesia, the region of Java and Sumatera are the most research locations (Figure 6). This is understandable since both islands are the most at risks from geophysical hazards.

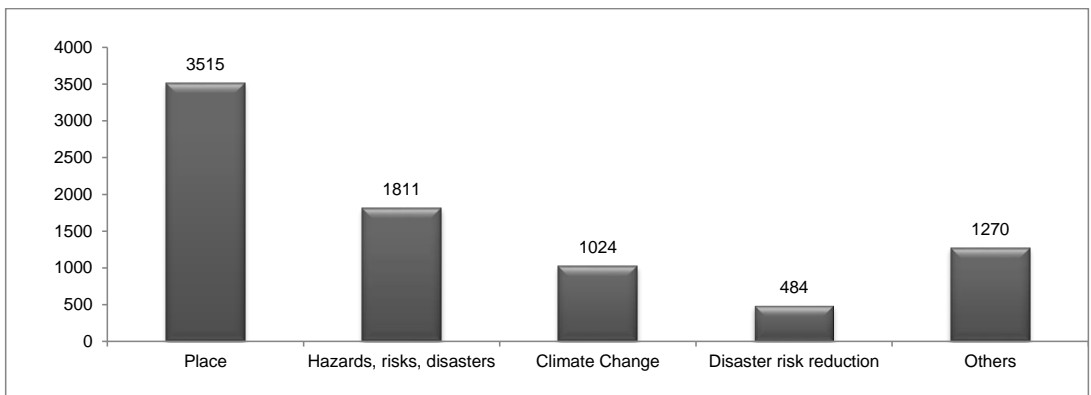


**Figure 5 categorization of keywords used**



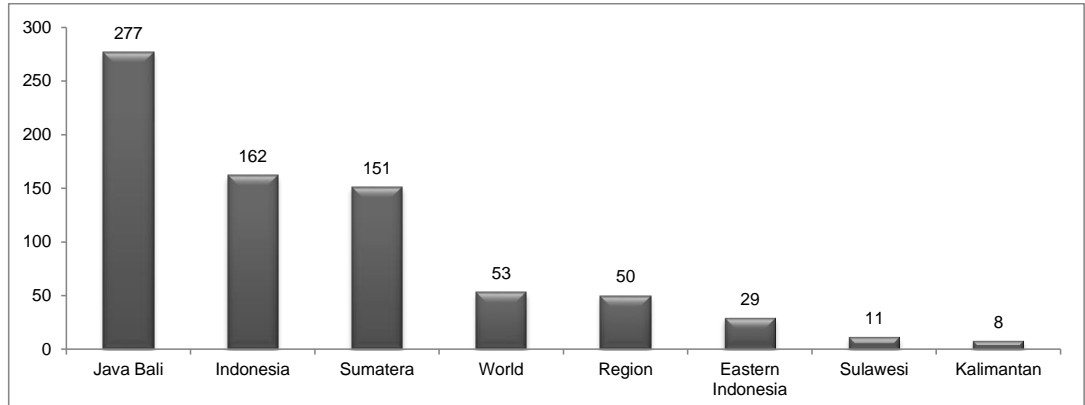

**Figure 6 Locations by which the researches are focused on (total 744)**

**3.1.4    Publications types**

This section presents where the publications are published. It is clear that publications from journal are those that got indexed the great majority, compared to conference proceedings, books, or others (Figure 7). A closer look on the journals shown that journals related to geophysical hazards (related to volcano, earthquake, tsunami, etc.) identification and assessments dominate the numbers of papers published on Indonesia (Table 6).


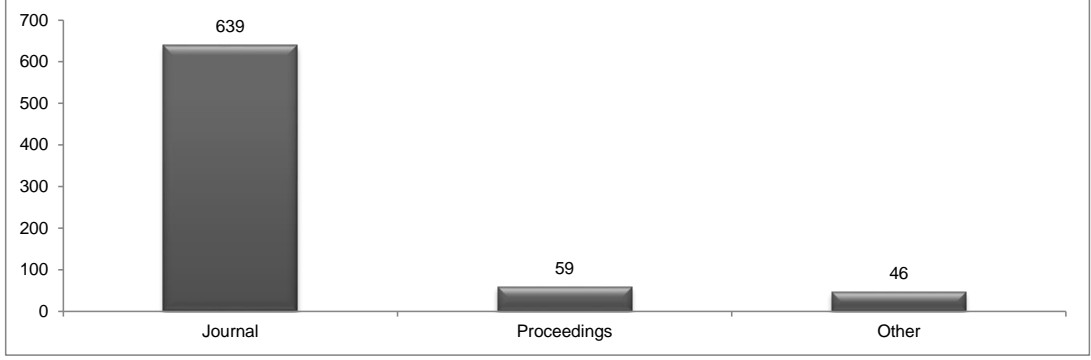

**Figure 7 Publications types (total 744)**

**Table 6 List of most frequent journals**

| Publications | Number of papers | IF / SJR | Category | | |
|---|---|---|---|---|---|
| | | | HRD | DRR | CC |
| Journal of Volcanology and Geothermal Research | 75 | 2.543 | x | | |
| Natural Hazards | 39 | 1.719 | x | x | |
| Natural Hazards and Earth System Science | 27 | 1.735 | x | x | |
| Bulletin of Volcanology | 22 | 2.519 | x | | |
| Geophysical Research Letters | 17 | 4.196 | x | | |
| Earth and Planetary Science Letters | 16 | 4.734 | x | | |
| Pure and Applied Geophysics | 15 | 1.618 | x | | |
| Nature | 14 | 41.456 | x | | x |
| Journal of Disaster Research | 14 | SJR 0.18 | | x | |





| Publications | Number of papers | IF / SJR | Category | | |
|---|---|---|---|---|---|
| | | | HRD | DRR | CC |
| Journal of Geophysical Research: Solid Earth | 12 | 3.426 | x | | |
| International Journal of Disaster Risk Reduction | 12 | SJR 0.510 | | x | x |
| Bulletin of the International Institute of Seismology and Earthquake Engineering | 12 | SJR 0.12 | x | | |
| Geomorphology | 11 | 2.785 | x | | |
| Disasters | 10 | 0.742 | | | |
| International Journal of Remote Sensing | 9 | 1.652 | x | | |
| Bulletin of the Seismological Society of America | 7 | 2.322 | x | | |


A very striking Figure, however, is shown when comparing the number of Indonesian journals that got indexed in SCOPUS as shown in Table 7. The Indonesian Journal of Geography is the only Indonesian journal included in the search with 7 papers listed. The journal was established in 1961 by the Faculty of Geography, of Gadjah Mada University, in cooperation with the Association of the Indonesian Geographers. The director of the editorial board is Sudibyakto, with Sartohadi,

Lavigne and Marfai as members of the editorial board (UGM, 2016). There are no clear counts on the number of academic journals in Indonesia, however, there are only 245 are accredited by DIKTI (Higher education directorates of the Ministry of Education) (DIKTI, 2016a) and 17 indexed in SCOPUS (DIKTI, 2016b). There are also none of these journals that have obtained an impact factor yet, and hence a Scientific Journal Ranking (SJR) Score is presented (SJR, 2016).

**Table 7 List of Indonesian Journals that are indexed with SCOPUS**

| Rank | Title | SJR | H index |
|---|---|---|---|
| 1 | Nutrition Bulletin | 0,267 | 25 |
| 2 | Bulletin of Chemical Reaction Engineering and Catalysis | 0,251 | 5 |
| 3 | Acta medica Indonesiana | 0,239 | 12 |
| 4 | Telkomnika | 0,236 | 6 |
| 5 | International Journal on Electrical Engineering and Informatics | 0,2 | 7 |
| 6 | Indonesian Journal of Chemistry | 0,175 | 2 |
| 7 | International Journal of Power Electronics and Drive Systems | 0,157 | 4 |
| 8 | Kukila | 0,152 | 2 |
| 9 | Journal of Engineering and Technological Sciences | 0,123 | 3 |
| 1 | Gadjah Mada International Journal of Business | 0,12 | 1 |
| 1 | Critical Care and Shock | 0,115 | 7 |
| 1 | International Journal of Technology | 0,115 | 2 |
| 1 | Journal of Mathematical and Fundamental Sciences | 0,112 | 4 |
| 1 | Journal of ICT Research and Applications | 0,106 | 2 |
| 1 | Biotropia | 0,103 | 1 |
| 1 | Indonesian Journal of Geography | | |
| 1 | Agrivita | 0,101 | 1 |

### 3.1.5    Key research topics

This section presents the more detailed findings of each of the three research topics.



### 3.1.5.1 Topics on hazards, risks and disasters assessments

The first sub-section explains the timelines, contents of researcher and locations inquired within or outside Indonesia on the
topic of hazards, risks and disasters assessments and identifications. As can be seen from Figure 8, there has been a gradual
increase on the number of published materials since 1978 to 1998. It is only in 2000 that the research in this topic reached its
first significant outputs of 25 publications. The next 4 years showed a sharp reduction in the number of publication. In 2004
the Indian Ocean tsunami occurred and hit Indonesian the most. Publications related the tsunami continued to be published
until it reached its peak in 2006. Then in 2009, the publications started to increase rapidly ever since and reached its peak in
2015 of 47 publications in a single year.

Most of the literature around this period focuses on the impacts of volcanic eruptions in Java and Sumatera. The oldest
publications related to hazards in Indonesia listed in Scopus are those by Neall (1976), Lahars as major geological Hazards
published in the Bulletin of the International Association of Engineering Geology, and one by Nakamura (1978) on the
Statistics of tsunamis in Indonesia in the Southeast Asian Studies. In terms of contributions by Indonesia researchers, the
earliest papers are by Sudradjat and Tilling (1984) on the Volcanic hazards in Indonesia: the 1982-83 eruption of
Galunggung, and Suryo and Clarke (1985) on the occurrence and mitigation of volcanic hazards in Indonesia as exemplified
at the Mount Merapi, Mount Kelud and Mount Galunggung volcanoes in the Quarterly Journal of Engineering Geology.

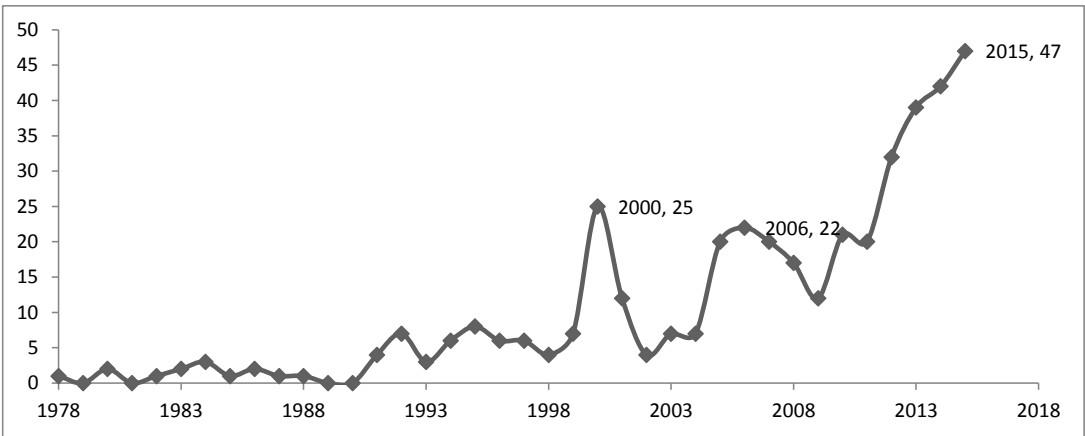

**Figure 8 Number of publications on hazards, risks, and disasters over the year (total 413)**

Utilizing the definition provided by EMDAT-CRED on the classifications of geophysical hazards, the study finds that there
are the majority of publications are related to volcanic eruption, dominated by the study of volcanoes in Java such as Merapi
(Andreastuti et al., 2000; Charbonnier and Gertisser, 2008; Gertisser et al., 2012; Lavigne, 1999; Verstappen, 1988; Voight
et al., 2000), Semeru (Carn, 1999; Siswowidjoyo et al., 1997; Solikhin et al., 2012; Thouret et al., 2007), Kelud (Lubis,
2014; Nakada et al., 2016)or Ijen (Heikens et al., 2005; Trunk and Bernard, 2008; van Hinsberg et al., 2010) (Figure 9). The
other hazard that receives many studies is related to examination of earthquake, how it happened, and methods to assess the
impacts. The research on tsunami receives gradual attention especially after 2004. There are also a small numbers of
publications related to landslide.






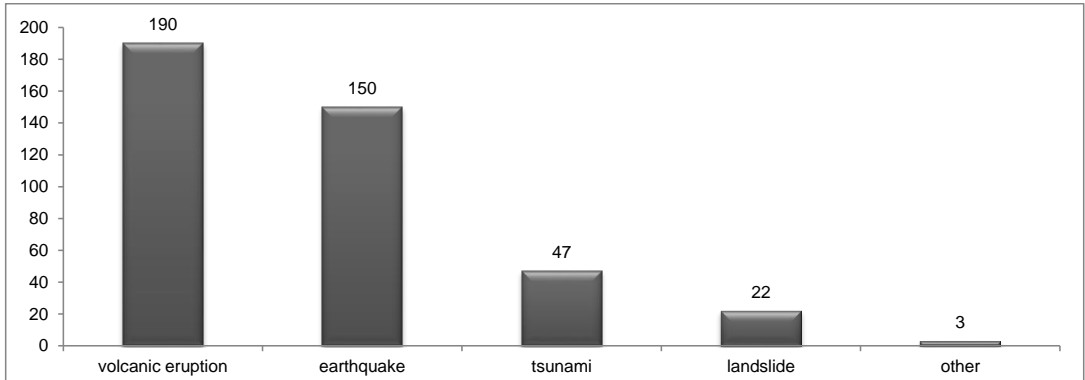

**Figure 9 Types of Hazards, risks, disasters (total 413)**

Figure 10 examines where these researches are focused worldwide, regionally or within Indonesia. As can be seed from the
Figure, the islands of Java and Sumatera are the two areas which receive examinations from the study. Correlating from the
previous figure, the study in these two islands is mostly correlated to the study of volcanic eruption, earthquake and tsunami.
This is not surprising considering that Indonesia has the most numbers of volcano, is located at the geographical ring or fire
where earthquakes occur the most, and also has experienced and been impacted by one of the most powerful earthquake of
8.9 R.S which caused tsunami in 2004 and hit Aceh, which is located in the island of Sumatera (Ishii *et al.*, 2005).


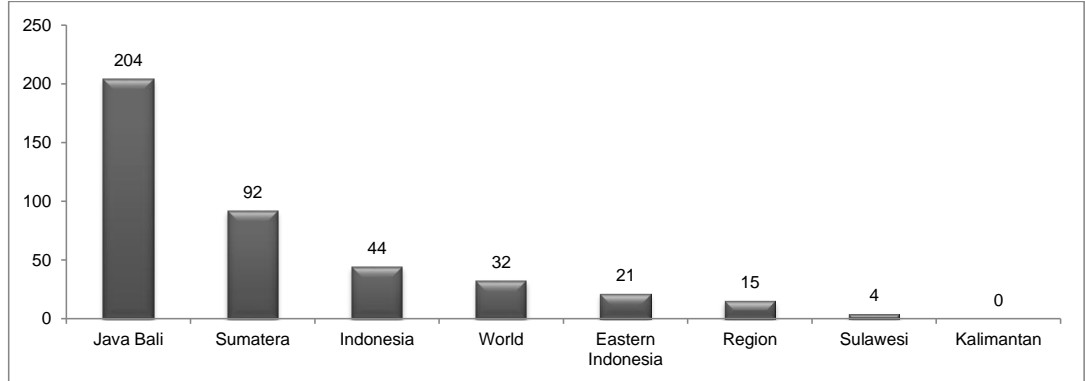

**Figure 10 Where the research focus is located (total 413)**

### 3.1.5.2    Topic on disaster risk reduction

The second sub-section explains the timelines, contents of researcher and locations inquired within or outside Indonesia on
the topic of disasters risk reduction. As can be seen from Figure 9, there have been very little publications published between
1978 and 2003. It is only after 2004 then there is a gradual increase of publications. The publication reach its peak in 2008,
after that it slightly reduced, and then continue to increase and reach another peak in 2013. Only then publications have
started to reduce. The oldest publications on DRR category is by Sudibyakto and Haroonah (1997) reviewing how disasters
are managed from a social science perspective in the Indonesian journal Geography.






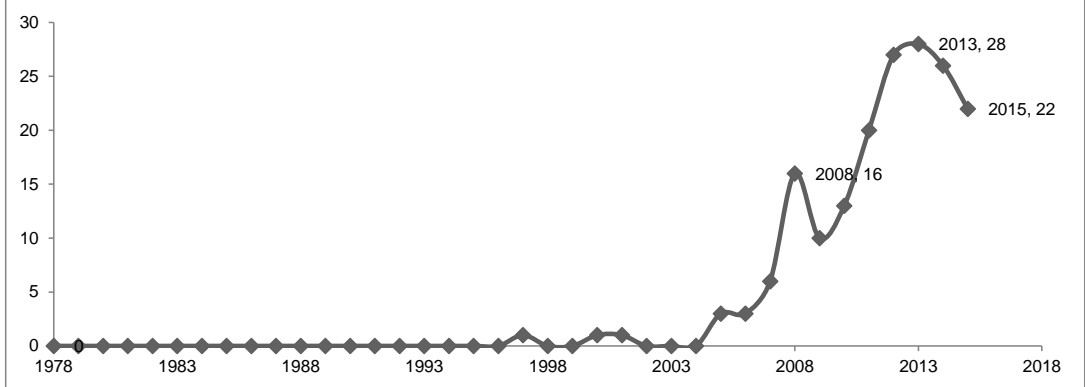

**Figure 11 Number of publications in DRR over the year (total 177)**

A close examination of 177 publications shows a very strong correlation between the issues discussed (Figure 12) and the locations (Figure 13). The topic that receives most attention in this category is related to the governance of DRR nationally
(Bakkour et al., 2015; Chang Seng, 2013; Djalante et al., 2013; Djalante et al., 2012; Guarnacci, 2012). The next topic that receives greater attention is on the evaluation of recover and reconstruction that have taken place after the 2004 Indian Ocean tsunami which hit Aceh, located in the Island of Sumatera (Chang et al., 2011; Daly and Brassard, 2011; Godavitarne et al., 2006; Guarnacci, 2012; Karan and Subbiah, 2011; Telford and Cosgrave, 2007). Within the period after 2004, other topics that are also related to the impacts of tsunami and disasters in general is the role of culture, gender, or religion in
helping the community to be resilient in facing disasters, and also how various disasters in Indonesia have impacted various community groups differently in relation to their culture or gender (Balgos et al., 2012; Baumann, 2008; Donovan, 2010; Donovan et al., 2012; Gaillard et al., 2008b; Guarnacci and Di Girolamo, 2012; Hiwasaki et al., 2015; Islam and Lim, 2015; Siagian et al., 2014).

Other topics that have been discussed were related to examination of early warning system especially in relation to tsunami
early warning system that has been one of the focuses of the Indonesian government to install them around Indonesia. One notable initiative was the German Indonesian Tsunami Early Warning Systems (GITEWS) (e.g. Schlurmann and Siebert, 2011; Steinmetz et al., 2010). There are also a large number of publications which examine the role of knowledge and information to help the community be more prepared to disasters (Dicky et al., 2015; Hiwasaki et al., 2015; Rafliana, 2012). There are 13 publications comparing Indonesia and Sri Lanka in regards the impacts of the tsunami on how it either become
the precursor for peace process in Indonesia but still take time for the process in Sri Lanka (Enia, 2008; Gaillard et al., 2008a; Hyndman, 2009; Kelman, 2005). Some lower numbers of papers examine community-based DRR which is strongly related to community preparedness (Adiyoso and Kanegae, 2013; Birkmann et al., 2015; Hidayati, 2012; James, 2008; Kusumasari and Alam, 2012), and others examine how children are affected psychologically from continuous exposures to hazards and disasters (Du et al., 2012; Lawler and Patel, 2012; Taylor and Peace, 2015; Vignato, 2012), and on emergency
management at the local or national level (Djalante et al., 2012; Esteban et al., 2013; Kusumasari and Alam, 2012).





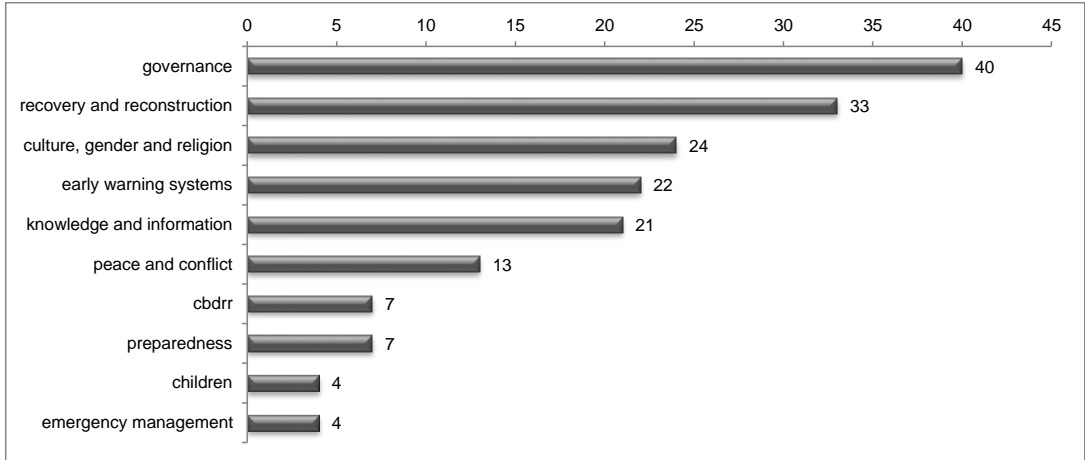

**Figure 12 Key issues discussed in DRR literatures (total 177)**

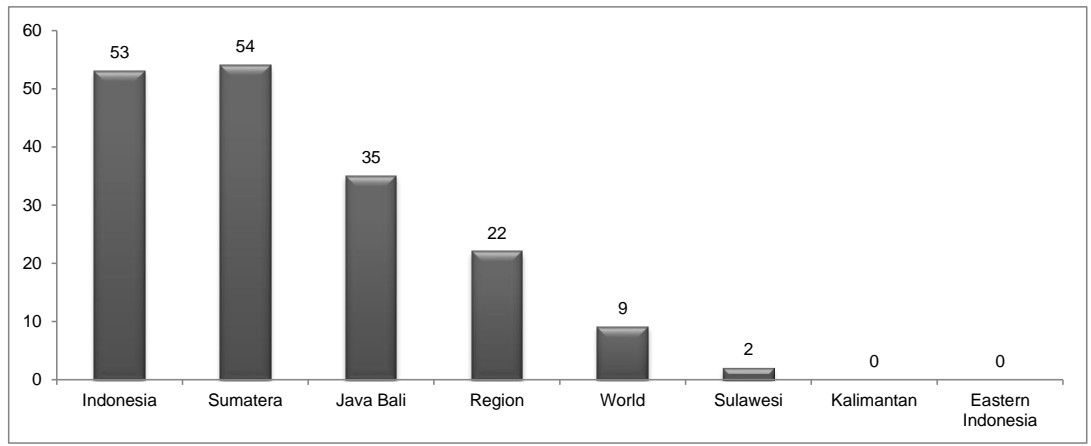


**Figure 13 Locus of the research (total 177)**

### 3.1.5.3    Topics on climate change

The third sub-section explains the timelines, contents of researcher and locations inquired within or outside Indonesia on the topic related to climate change. As can be seen from Figure 14, there has been few publications within the period between 1978 and 1990. The second period between 1990 to 2000 see a slight increase in literature, then there were 5 literatures published in 2001. These are related to examinations on the causes and impacts of the forest fires in Indonesia. The numbers of publications did not change in general until 2008. It is only after 2010 that there is a sharp increase in the numbers of publications and reach its peak in 2015 of 35 papers. The earliest publication was in 1992 by Subijakto (1992) who examine the facts and future trends of climate change: a case study of the eastern part of the Indonesia islands.  Other paper that examine the management of climate change impacts in Indonesia is written by Murdiyarso (1993), in the Chemosphere Journal on the Policy options to reduce $CO^2$ release resulting from deforestation and biomass in Indonesia.







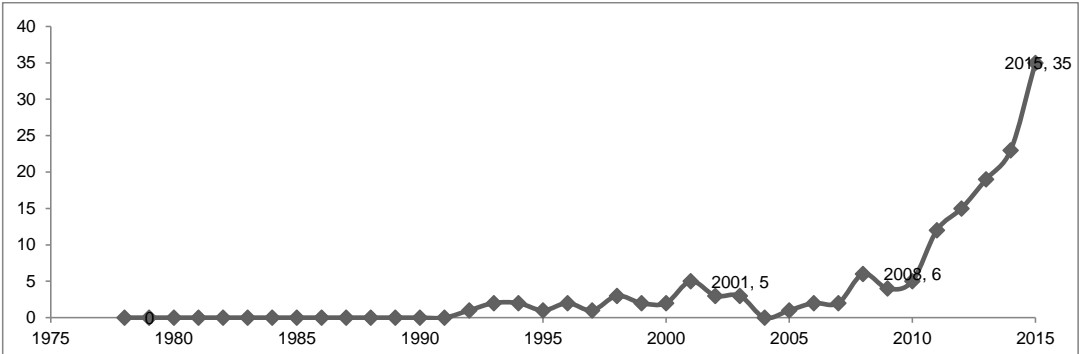

**Figure 14 Number of publications on the topic of climate change (total 154)**


As shown in Figure 15, the author categorizes the 154 publications in this group into three major discussions related to the impacts of climate change on Indonesia, on the governance of climate change adaptation, and also on there is a significant numbers of publications related to the issues of deforestation and land degradation which has taken enormous impacts on Indonesian rain forest. Indonesia is one of the countries that house some of the largest coverage of rainforest in the world

especially in the islands of Sumatera and Kalimantan.

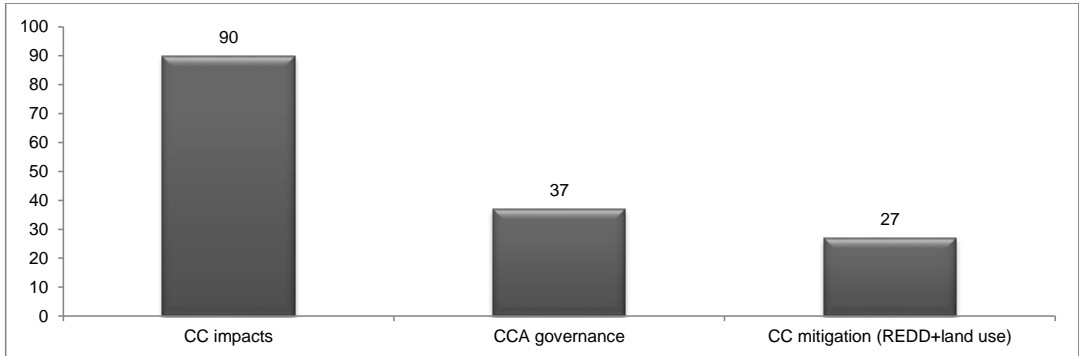

**Figure 15 Key issues discussed in climate change literature (total 154)**

Since the majority of materials published in this category are related to the review on the impacts on climate change in Indonesia, this paper examines deeper on those literature (Figure 16). It is shown that the impact on crops production, mainly on rice has been the majority of those researches (Caruso et al., 2016; D'Arrigo et al., 2011; D'Arrigo and Wilson, 2008; Kawanishi and Mimura, 2015; Keil et al., 2009; Naylor et al., 2001; Sano et al., 2013; Shofiyati et al., 2014). This is strongly related to the examination of too much water which can cause flood (Marfai and King, 2008a; Marfai et al., 2008; Marfai et

al., 2014, 2015; Muis et al., 2015; Neolaka, 2012, 2013; Sarminingsih et al., 2014; Shrestha et al., 2014)or too little water which can or have caused drought in Indonesia (Aldrian and Djamil, 2008; D'Arrigo and Smerdon, 2008; D'Arrigo and Wilson, 2008; D'Arrigo et al., 2006; Keil et al., 2009; Keil et al., 2008). A high number of researches are also done on linking droughts (D'Arrigo and Smerdon, 2008; D'Arrigo et al., 2006; Salafsky, 1994; Shofiyati et al., 2014) and fire (Brauer and Hisham-Hashim, 1998; Fang and Huang, 1998; Jim, 1999; Page et al., 2002; Stolle and Lambin, 2003; Stolle and

Tomich, 1999; Usman and Hartono, 1997) occurrences especially those on forest fire. There are also research on sea level rise and its impacts on coastal areas. A small number of research focuses on temperature, rainfall (Aldrian and Djamil, 2008;




Chrastansky and Rotstayn, 2012; D'Arrigo and Wilson, 2008). The impact on health (Coughlan de Perez *et al.*, 2015) and animal (Morwood et al., 2008; Purnomo et al., 2011) has also received some attention.

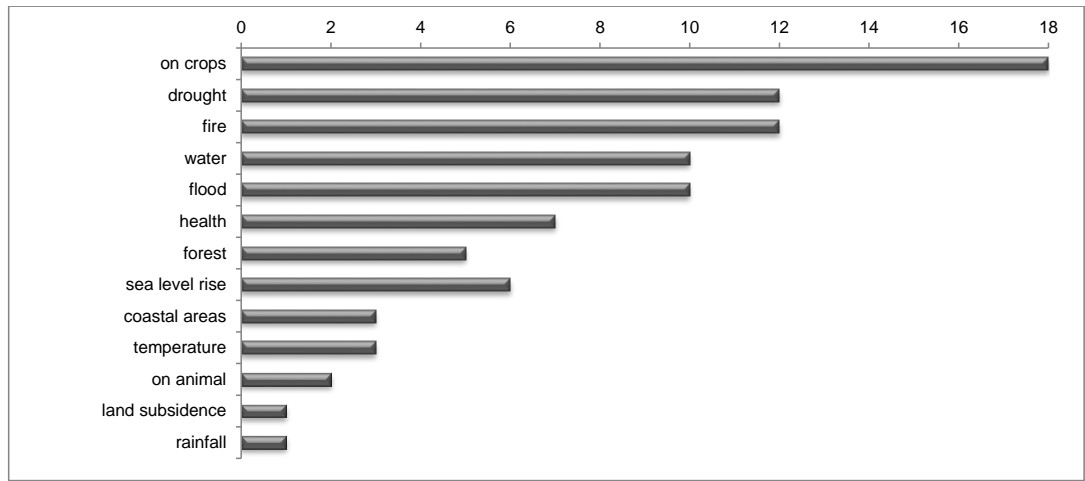


**Figure 16 Key issues discussed in impacts of climate change (total 90)**

In relation the area by which this research is located, the islands of Sumatera and Java has become the two major locations on the research of the impacts since they are the area where greatest paddy fields and crops productions are located (Figure 360 17). It is also great to see that various locations in Indonesia such as those in Sulawesi and also eastern part of Indonesia have received examinations in some of those studies.

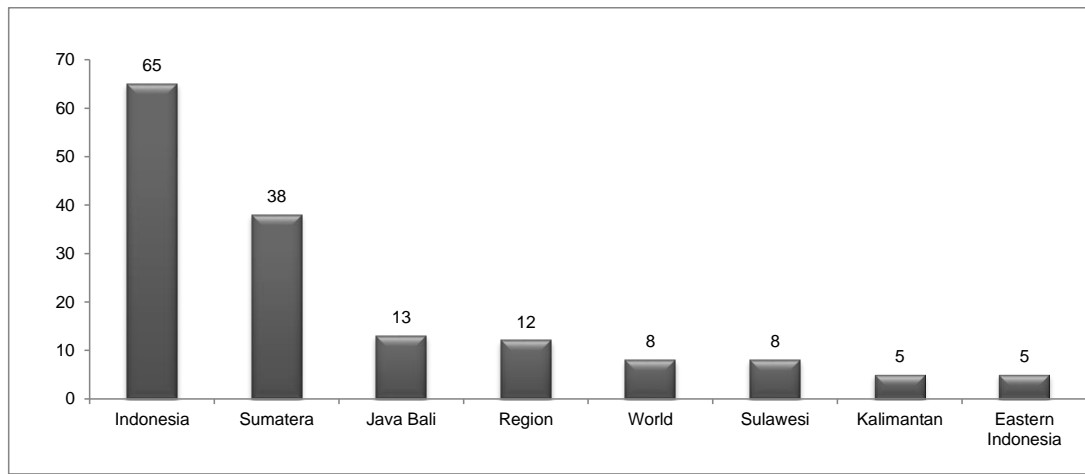

**Figure 17 Where the research focus is located (total 154)**

**3.2 Reviewing progress of Indonesian researchers and organizations**

This section examines the roles of Indonesian researchers and Indonesian organizations in contributing the production of those literatures, and also on how the Indonesian researchers have been in collaborating with other International / non-Indonesian organizations, and also in producing high quality papers.





### 3.2.1 Authorships

This study examines authorships of the publications in terms who published the most numbers of papers, and how Indonesian authors have been progressing in terms of publications. In general, the contribution of international / non-Indonesian authors dominates the productions of publications. The comparison shows that there are 2 international authors for every Indonesian author. Figure 18 shows there are more than double the number of international compared to Indonesians authors but more than half of the publications are co-authored by at least one Indonesian. In more detail, Figure

19 shows the comparison of involvement of Indonesian authors in the three major groups of publications. It can be seen that there are more authors in the hazards, risks and disaster assessments group, and the rest is divided almost equally by those in the other two groups.

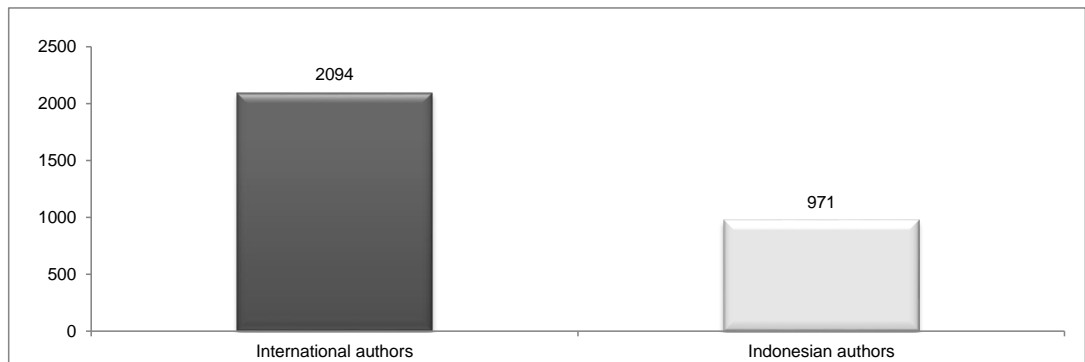

**Figure 18 Comparing the numbers of Indonesian and non-Indonesian authors (total=3065)**

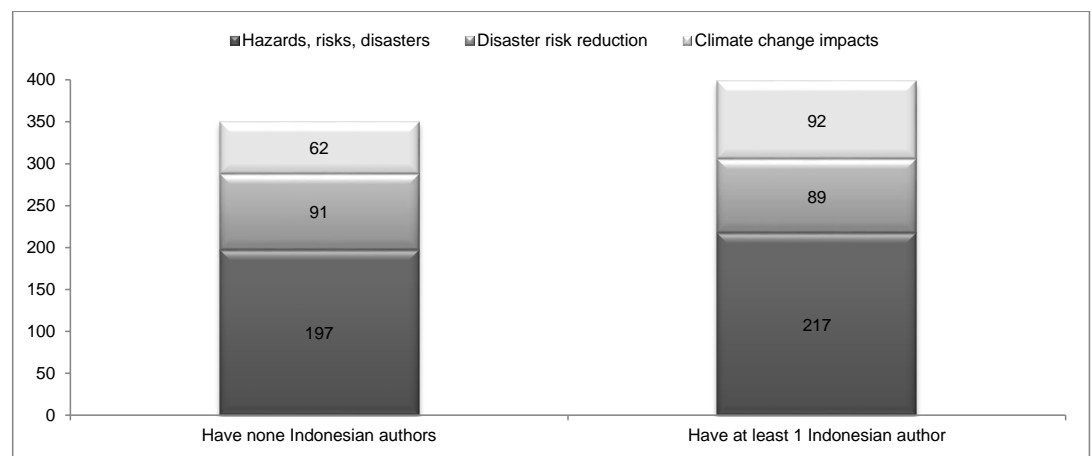

**Figure 19 Comparing the number of Indonesian and non-Indonesian authors with 3 key research topics (Total 744)**

Table 8 shows the list of top ten authors with highest number of publications. Frank Lavigne from Universite Paris 1 Pantheon Sorbonne published the highest numbers of papers (Google Scholar, 2016d). Lavigne worked closely with JC Thouret from Laboratory Magmas et Volcanis, who is in the 4[th] list (LMV, 2016). Gertisser is a senior lecturer in Keele University (Google Scholar, 2016f). Voight is a renowned geologist and volcanologist in USA who have worked on the Mount Merapi since 1980s (Google Scholar, 2016b). Sieh has long collaborated with Natawidjaja on their works on

seismology in Indonesia (EOS, 2016). Surono and Hendrasto are both affiliated with the PVMBG (PVMBG, 2016). Marfai





is affiliated at the Gadjah Mada University who has written on the topic of DRR, and also examination of hydro-meteorological hazards and disasters (Google Scholar, 2016a).

**Table 8 List of top ten authors with highest number of publications**

| Author | Organization / Country | Number of publication in this study | SCOPUS profile (profile, citations, h-index, most frequent collaborator) | Google Scholar profile (publications, citations, h-index, i10-index) | Research Gate profile (publications, citations, impact points) | Other profile |
|---|---|---|---|---|---|---|
| Lavigne, Frank | France / Universite Paris 1 Pantheon Sorbonne | 28 | 62, 1152, 19, more than 150, | 124, 1648, 21, 34 | 153, 1,430, 162.61 | |
| Surono | Indonesia / PVMBG (Volcanology Survey Indonesia) | 18 | 27, 348, 12, 125, | NA | NA | https://en.wikipedia.org/wiki/Surono_(volcanologist) |
| Abidin, Hasanuddin Zainal | Indonesia / Institute Teknologi Bandung (ITB) | 16 | 53, 493, 11, 121, Andreas H | 172, 1513, 19, 35 | NA | http://www.fitb.itb.ac.id/en/hasanuddin.abidin/ |
| Thouret, Jean-Claude | France / Laboratory Magmas er Volcanis | 16 | 114, 1147, 20, More than 150,Gourgaud, A | NA | NA | http://pendientedemigracion.ucm.es/info/agr/particip/cv/thouret.html# |
| Gertisser, Ralf | United Kingdom / Keele University | 15 | 42,684,468,14, aboce150,Charbonnier SJ | NA | 87 803 132,51 | https://www.keele.ac.uk/gge/people/ralfgertisser/ |
| Voight, Barry | USA / Pennsylvania State University | 14 | | NA | 250 5,307 570.75 | |
| Sieh, Kerry. | Singapore / Earth Observatory of Singapore | 13 | 120, 5752, 43, more than150, Natawidjaja, DH | NA | NA | http://www.earthobservatory.sg/people/kerry-sieh |
| Natawidjaja, Danny Hilman | Indonesia / LIPI | 11 | 42, 1913, 21,123, Sieh KE | 147, 2964, 25, 33 | NA | |
| Marfai, Muhammad Aris | Indonesia / UGM | 11 | 19, 183, 8, 36, King, L | 79, 517, 12, 14 | NA | http://arismarfai.staff.ugm.ac.id/main/?page_id=44 |
| Hendrasto, Muhammad | Indonesia / PVMBG (Volcanology Survey Indonesia) | 10 | 16, 92, 4, 59, Surono | NA | NA | NA |


Table 9 lists the Indonesian authors with 10 highest publications. The highest publications finally selected for the review of an Indonesian author is 18 publications by Surono of PVMBG. Abidin of the ITB has been listed to have 16 publications in this review, while his Google scholar profile shows that he has published extensively of 172, and with 1513 citations (Google Scholar, 2016e). There are a limited numbers of authors had been involved with publications to the highest IF journals such as Nature and Science. One of these authors is Natawidjaja who has 147 publications with total citations of 2964 based on his Google Scholar profile (Google Scholar, 2016c).


**Table 9 List of Indonesian authors with top ten publications**

| Indonesian Author | No. of publications in this review | Org. | SCOPUS profile (publications, citations, h-index, co-authors, most frequent collaborator) | Google Scholar profile (publications, citations, h-index,i-10 index) |
|---|---|---|---|---|
| Surono, | 18 | PVMBG | 27, 348, 12, 125, Hendrasto M | NA |
| Abidin, H.Z. | 16 | ITB | 53, 493, 11, 121, Andreas H | NA |





| Natawidjaja, D.H. | 11 | LIPI | 42, 1913, 21, 123, Sieh KE | 147, 2964, 25, 33 |
|---|---|---|---|---|
| Marfai, M.A. | 11 | UGM | 19, 183, 8, 36, King, Lorenz | 79, 517, 12, 14 |
| Hendrasto M | 10 | PVMBG | 16, 92, 4, Surono | NA |
| Andreas, H. | 10 | ITB | 20, 123, 6, 46, Abidin, H Z | NA |
| Ratdomopurbo, A. | 8 | NTU | 17, 441, 10, 59, Lühr, B G | NA |
| Muhari, A. | 8 | MAAF | 15, 112, 6, 53, Imamura, F | NA |
| Sumarti, Sri. | 8 | UGM | 14, 367, 13, 84, Surono | NA |
| Suwargadi, BW | 7 | LIPI | 31, 1102, 17, 103, Natawidjaja, DH | 97, 1585, 20, 24 |

Figure 20 shows the distribution of Indonesian authors who have more than 1 publication selected in this review (see appendix 1 for full list of authors). There are 21 organizations located in Java and Bali, dominated by ITB and UGM, and there is 1 in Kalimantan, the University of Syiah Kuala (Aceh) and the University of Mataram (Lombok Island). There are 18 national level organizations such as LIPI, PVMBG, LAPAN, BMKG, Bakosurtanal, while 6 Indonesians are currently working outside Indonesia. In terms of distribution of males and females, the composition is almost 4 to 1. There only 15

who has Goggle Scholar or Research Gate profiles.

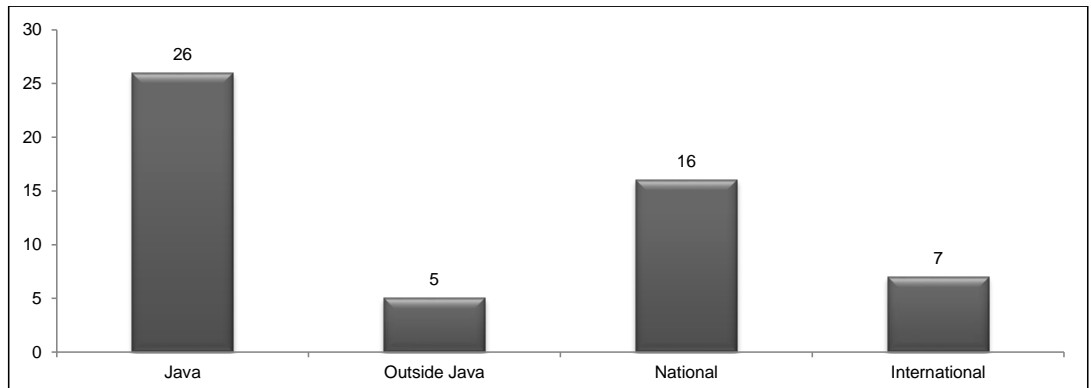

**Figure 20 Composition of Indonesian authors with publications more than 1**

This results show a great deal on the need for increasing the capacity of Indonesian authors to have the skills and experiences

in writing in English and submit for internationally regarded journal publications. Indonesian authors largely lack the experience in international collaborations and the language and writing skills necessary for submitting their works into internationally accredited journals. Despite some Indonesian researchers who have been strongly influential within the study of hazards, DRR or climate change in Indonesia and could potentially contribute to the global development of knowledge in these fields, they only published in Bahasa Indonesia and did not submit their works into international English written

journals.

### 3.2.2    Research centers/organizations affiliations

This section examines the place and organizations by which the researchers are affiliated, systematically from the regional, to national, and amongst organizations in Indonesia. The organizations, which house ten most productive publications related to this review, are shown in Figure 21. In general, there are equal number of organizations that are based in

Indonesia, and their contributions is comprised slightly more than half the overall contributions amongst these most productive agencies.





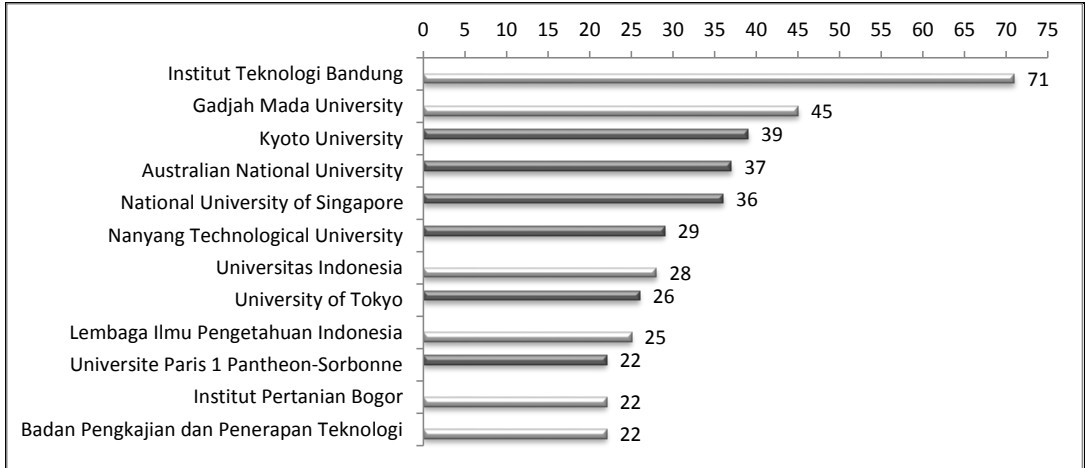

**Figure 21 Organizations with highest number of publications**

Figure 22 looks deeper on the contribution of different organizations within Indonesia. It is shown that ITB and UGM
dominate almost half the total publications. There are also more twice universities in Java that those outside Java, while the
rest of publications are contributed by national level organizations such as LIPI and PVMBG.

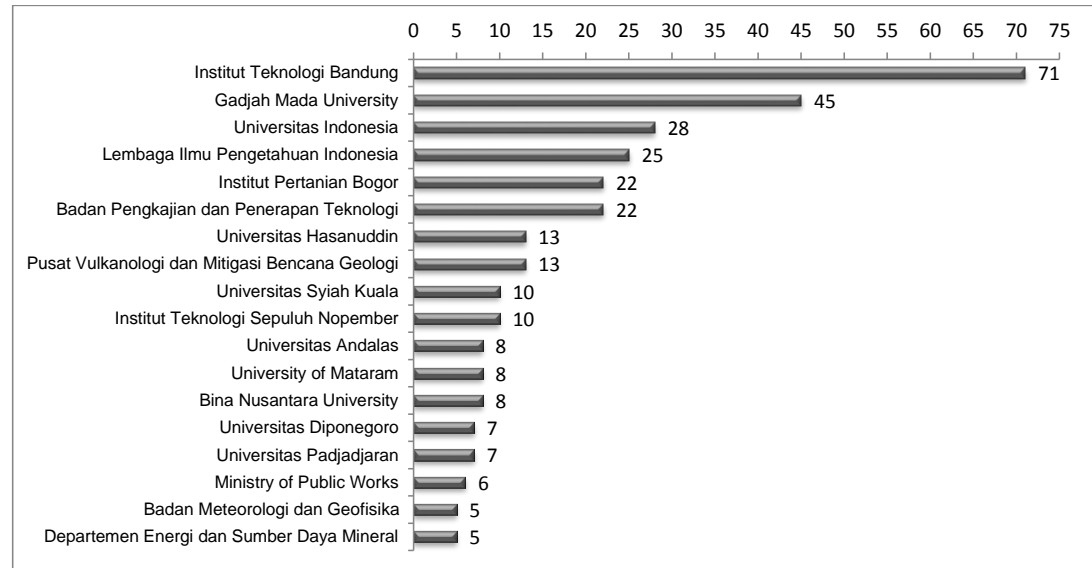

**Figure 22 Contributions of Indonesian organizations**

### 3.2.3   Research quality measured through journal impact factor and number of citation

This section presents the research quality of the publications, measured through the journal impact factors and the number of
citations. Most importantly, it evaluates the progress of the Indonesian scholars through comparing their research outputs
between papers first authored by Indonesian and overall papers. It does so through comparing the overall progress, and
through examination of each research topic group.





### 3.2.3.1 Overall

This section the list of 10 most cited publications through comparing the roles of those published in general by any authors, and those publications that are published by an Indonesia first author. Figure 23 shows the comparison between the progress of Indonesian researchers in 10 most cited papers overall and those first authored by Indonesian. Important observations are that there are more authors in 10 most cited papers, more international authors in most 10 cited papers, more Indonesians in 10 cited paper first authored by Indonesian, 10 most cited papers have higher impact factor, and 10 most cited papers have

higher citations. This might suggest that Indonesians researchers tend to work with other Indonesians and hence needed to expand their collaborations with international scholars as a strategy to increase the number of citations and ability to submit for higher impact journals.

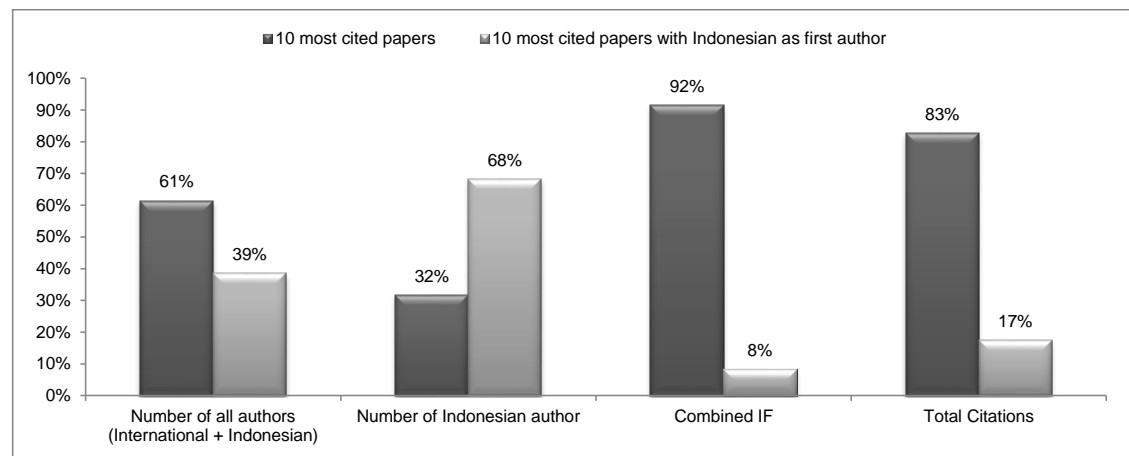

**Figure 23 comparing the roles of Indonesian researchers in the 10 most cited papers**


Table 10 shows the list of 10 most cited papers of all publications. With the 10 most cited papers, the total citations is 3,427 with combined impact factor (IF) is 256.013, and there are only 32% of the authors are Indonesian, and none of them are first authors. It is shown that they are published in high impact factor journals such as Nature, Science, or those related to geophysical hazards. The two highest cited papers are published in Nature Journal and discussed the impacts on the forest

fires in Indonesia. The paper related to the examination of the amount of carbon released from peat and forest fires in Indonesia in 1997 has the highest citation of 1156 by Page et al (2002), published in Nature. The majority of the paper discussed major hazards from earthquake in Sumatera (Briggs et al., 2006; Hsu et al., 2006; Ishii et al., 2005; Konca et al., 2008), and the rest review the impacts of Toba (Rampino and Self, 1992) and Merapi volcanic eruption (Voight et al., 2000).

There are 6 papers which also have Indonesians to contribute. Jaya and Limin are both lecturers from the Palangkaraya University in Kalimantan, where this forest fire occurred across the rain forest and impacted not only Indonesia but also the countries in the region such as Singapore (Tay, 1998) and Malaysia (Khandekar *et al.*, 2000). Subarya, Natawidjaja, along with Sieh contributed the most (Briggs et al., 2006; Hill et al., 2012; Horspool et al., 2014; Hsu et al., 2006; Konca et al., 2008; Muhari et al., 2010; Nalbant et al., 2005; Philibosian et al., 2012; Prayoedhie et al., 2012; Schlurmann et al., 2010;

Singh et al., 2010).

A closer examination on the list of ten most cited publications with Indonesian as first author shows a very striking picture. Table 11 shows the list of 10 most cited papers by Indonesian as first author. The total citations is only 720, with combined IF of only 23, 492, with 80% of the authors are Indonesian. The papers are much more varied in terms of topics they



discussed. There is no single paper in this Table that become the 10 most cited paper overall. The first two most cited papers
are related to impacts of climate change in Indonesia. Aldrian (2003), Susanto (2003; 2001) and Amien et al (1996) authored
papers related to climate change or its impacts on Indonesia. There are papers that examine impacts of volcano (Andreastuti
et al., 2000; Widiwijayanti et al., 2009), earthquake (Irsyam *et al.*, 2008) and tsunami (Muhari *et al.*, 2011), while the rest
examine land subsidence in Jakarta (Abidin *et al.*, 2011) and progress of DRR governance nationally (Djalante et al., 2012).


**Table 10 List of 10 most cited papers, comparing overall papers and those first authored by Indonesian**

| Title | Year | Journal | No. of citations | All Authors | Indonesian author(s) | Impact factor | Title | Year | Journal Name | No of citations | Authors | Indonesian author(s) | IF |
|---|---|---|---|---|---|---|---|---|---|---|---|---|---|
| The amount of carbon released from peat and forest fires in Indonesia during 1997 | 2002 | Nature | 1156 | Page S.E., Siegert F., Rieley J.O., Boehm H.-D.V., Jaya A., Limin S. | Adi Jaya and Suwido Limin (University of Palangkaraya, Kalimantan) | 41.456 | Identification of three dominant rainfall regions within Indonesia and their relationship to sea surface temperature | 2003 | International Journal of Climatology | 282 | Aldrian E., Dwi Susanto R. | Aldrian E., Dwi Susanto R. | 3.157 |
| Increased damage from fires in logged forests during droughts caused by El Niño | 2001 | Nature | 473 | Siegert F., Ruecker G., Hinrichs A., Hoffmann A.A. | - | 41.456 | Upwelling along the coasts of Java and Sumatra and its relation to ENSO | 2001 | Geophysical Research Letters | 137 | Susanto R.D., Gordon A.L., Zheng Q. | Susanto R.D., | 4.196 |
| Extent, duration and speed of the 2004 Sumatra-Andaman earthquake imaged by the Hi-Net array | 2005 | Nature | 355 | Ishii M., Shearer P.M., Houston H., Vidale J.E. | - | 41.456 | Effects of interannual climate variability and climate change on rice yield in Java, Indonesia | 1996 | Water, Air, and Soil Pollution | 46 | Amien I., Rejekiningrum P., Pramudia A., Susanti E. | Amien I., Rejekiningrum P., Pramudia A., Susanti E. | 1.554 |
| Plate-boundary deformation associated with the great Sumatra-Andaman earthquake | 2006 | Nature | 326 | Subarya, Chlieh, Prawirodirdjo, Avouac, Bock, Sieh, Meltzner, Natawidjaja, McCaffrey | Subarya, Prawirodirdjo, Natawidjaja, | 41.456 | A detailed tephrostratigraphic framework at Merapi Volcano, Central Java, Indonesia: Implications for eruption predictions and hazard assessment | 2000 | Journal of Volcanology and Geothermal Research | 67 | Andreastuti S.D., Alloway B.V., Smith I.E.M. | Andreastuti S.D., | 2.543 |


| Title | Year | Journal Name | No of citations | Authors | Indonesian author(s) | IF |
|---|---|---|---|---|---|---|
| Land subsidence of Jakarta (Indonesia) and its relation with urban development | 2011 | Natural Hazards | 49 | Abidin H.Z., Andreas H., Gumilar I., Fukuda Y., Pohan Y.E., Deguchi T. | Abidin H.Z., Andreas H., Gumilar I., Pohan Y.E., | 1.719 |
| Building resilience to natural hazards in Indonesia: Progress and challenges in implementing the Hyogo Framework for Action | 2012 | Natural Hazards | 30 | Djalante R., Thomalla F., Sinapoy M.S., Carnegie M. | Djalante R., Sinapoy M.S., | 1.719 |
| The role of fire in changing land use and livelihoods in Riau-Sumatra | 2004 | Ecology and Society | 29 | Suyanto S., Applegate G., Permana R.P., Khususiyah N., Kurniawan I. | Suyanto S., Permana R.P., Khususiyah N., Kurniawan I. | 3.310 |
| Examination of three practical run-up models for assessing tsunami impact on highly populated areas | 2011 | Natural Hazards and Earth System Science | 27 | Muhari A., Imamura F., Koshimura S., Post J. | Muhari A., | 1.735 |

| Title | Year | Journal | No. of citations | All Authors | Indonesian author(s) | Impact factor |
|---|---|---|---|---|---|---|
| Volcanic winter and accelerated glaciation following the Toba super-eruption | 1992 | Nature | 307 | Rampino M.R., Self S. | - | 41.456 |
| Neotectonics of the Sumatran fault, Indonesia | 2000 | Journal of Geophysical Research: Solid Earth | 281 | Sieh, Natawidjaja | Danny Natawidjaja | 3.426 |
| Frictional afterslip following the 2005 Nias-Simeulue earthquake, Sumatra | 2006 | Science | 246 | Hsu Y.-J., Simons M., Avouac J.-P., Galeteka J., Sieh K., Chlieh M., Natawidjaja D., Prawirodirdjo L., Bock Y. | Natawidjaja D., Prawirodirdjo L (LIPI) | 33.61 |
| Deformation and slip along the Sunda megathrust in the great 2005 Nias-Simeulue earthquake | 2006 | Science | 211 | Briggs R.W., Sieh K., Meltzner A.J., Natawidjaja D., Galetzka J., Suwargadi B., Hsu Y.-J., Simons M., Hananto N., Suprihanto I., Prayudi D., | Natawidjaja D., Suwargadi B Hananto N., Suprihanto I., Prayudi D., Prawirodirdjo L (LIPI) | 33.61 |


| Title | Year | Journal | No. of citations | All Authors | Indonesian author(s) | Impact factor |
|---|---|---|---|---|---|---|
| | | | | Avouac J.-P., Prawirodirdjo L., Bock Y. | | |
| Partial rupture of a locked patch of the Sumatra megathrust during the 2007 earthquake sequence | 2008 | Nature | 189 | Konca A.O., Avouac J.-P., Sladen A., Meltzner A.J., Sieh K., Fang P., Li Z., Galetzka J., Genrich J., Chlieh M., Natawidjaja D.H., Bock Y., Fielding E.J., Ji C., Helmberger D.V. | Natawidjaja D.H. (LIPI) | 41.456 |
| Historical eruptions of Merapi Volcano, Central Java, Indonesia, 1768-1998 | 2000 | Journal of Volcanology and Geothermal Research | 167 | Voight B., Constantine E.K., Siswowidjoyo S., Torley R. | Siswowidjoyo S., (PVMBG) | 2.543 |
| | | | Total citations: 3427 | Total number of authors: 75 | Total number of non-Indonesian authors: 13 | Total IF: 256.013 |

| Title | Year | Journal Name | No of citations | Authors | Indonesian author(s) | IF |
|---|---|---|---|---|---|---|
| Objective rapid delineation of areas at risk from block-and-ash pyroclastic flows and surges | 2009 | Bulletin of Volcanology | 27 | Widiwijayanti C., Voight B., Hidayat D., Schilling S.P. | Widiwijayanti C., Hidayat D., | 2.519 |
| Proposed seismic hazard maps of Sumatra and Java islands and microzonation study of Jakarta city, Indonesia | 2008 | Journal of Earth System Science | 26 | Irsyam M., Dangkua D.T., Hendriyawan, Hoedajanto D., Hutapea B.M., Kertapati E.K., Boen T., Petersen M.D. | Irsyam M., Dangkua D.T., Hendriyawan, Hoedajanto D., Hutapea B.M., Kertapati E.K., Boen T., | 1.040 |
| | | | Total citations: 720 | Total number of authors: 47 | Total number of Indonesian authors: 28 | Total IF: 23.492 |





### 3.2.3.2 Topics on hazards, risks, and disasters assessments

This sub-section examines the roles of Indonesian researchers in the publications of materials related to the topic of hazards, risks and disasters assessments. As can be seen from Figure 24, there are twice the number of authors in the most cited

papers than the Indonesians, while the opposite occurs in when comparing the 10 cited papers with Indonesian as first author.

This shows that Indonesian authors needed to collaborate more with international authors. A very striking Figure is shown when examining the impacts of the publications. The total combined impact factor and citations are more than 9 times greater when comparing 10 most cited papers in general to those first authored by Indonesians. This shows how lacking the

quality and impact of publications written by Indonesian scholars in general. Nature and Science are the two most frequent journals while Natawidjaja and Siswowidjoyo are the only 2 Indonesian authors in the 10 most cited papers overall. There are Indonesians in every paper that is first authored by an Indonesian. Those from ITB and UGM dominate the list.

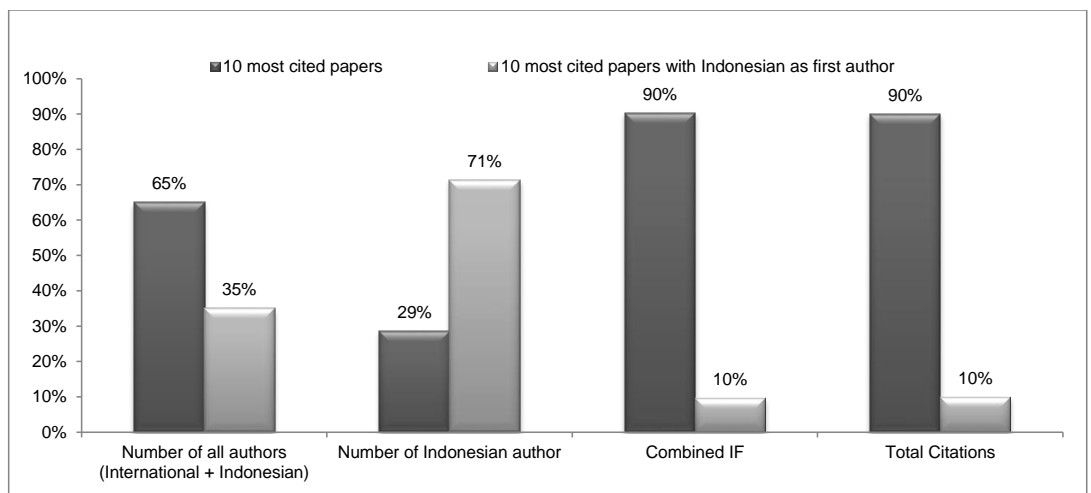

**Figure 24 Comparing the role of Indonesian authors in 10 most cited publications in HRD categories**

Table 11 lists ten most cited papers in the category of hazards, risks and disaster assessments. There are six papers examine earthquake in Sumatera, and two talks about the impact of Toba super-eruption, and the rest talk about Merapi volcano. The highest cited paper is that of Ishii M et al (2005) on the extent their examination of the 2004 Sumatran-Andaman earthquake

which caused the Indian Ocean tsunami. Along with Professor Natawidjaja, Professor Kerry Sieh has been involved in three of these most cited papers (Briggs et al., 2006; Hsu et al., 2006).

Table 11 shows the list of 10 most cited papers in this category with Indonesian as first author. Four papers discuss volcano, two papers discuss earthquake, and the rest discuss tsunami and landslide risks. One particular finding here is that there are 4

papers by which all authors are all Indonesians which also suggest that Indonesians researchers might still tend to work with other Indonesians and hence needed to expand their collaborations with international scholars as a strategy to increase the number of citations and ability to submit for higher impact journals. Andreastuti (2000) has the highest cited paper on the works on Merapi volcano, followed by the work of Abidin (2011) on land subsidence in Jakarta.



**Table 11 List of 10 most cited paper in the hazards, risks and disasters categories**

| Title | Year | Journal Name | No. of citation | Authors | Indonesian author(s) | Impact Factor |
|---|---|---|---|---|---|---|
| Extent, duration and speed of the 2004 Sumatra-Andaman earthquake imaged by the Hi-Net array | 2005 | Nature | 355 | Ishii M., Shearer P.M., Houston H., Vidale J.E. | | 41.456 |
| Volcanic winter and accelerated glaciation following the Toba super-eruption | 1992 | Nature | 307 | Rampino M.R., Self S. | | 41.456 |
| Frictional afterslip following the 2005 Nias-Simeulue earthquake, Sumatra | 2006 | Science | 246 | Hsu Y.-J., Simons M., Avouac J.-P., Galeteka J., Sieh K., Chlieh M., Natawidjaja D., Prawirodirdjo L., Bock Y. | Natawidjaja D., Prawirodirdjo L., (LIPI) | 33.61 |
| Deformation and slip along the Sunda megathrust in the great 2005 Nias-Simeulue earthquake | 2006 | Science | 211 | Briggs R.W., Sieh K., Meltzner A.J., Natawidjaja D., Galetzka J., Suwargadi B., Hsu Y.-J., Simons M., Hananto N., Suprihanto I., Prayudi D., Avouac J.-P., Prawirodirdjo L., Bock Y. | Natawidjaja D., Suwargadi B., Hananto N., Suprihanto I., Prayudi D., Prawirodirdjo (LIPI) | 33.61 |
| Partial rupture of a locked patch of the Sumatra megathrust | 2008 | Nature | 189 | Konca A.O., Avouac J.-P., Sladen A., Meltzner A.J. | Natawidjaja D.H., (LIPI) | 41.456 |
| A detailed tephrostratigraphic framework at Merapi Volcano, Central Java, Indonesia: Implications for eruption predictions and hazard assessment | 2010 | Journal of Volcanology and Geothermal Research | 67 | Andreastuti S.D., Alloway B.V., Smith I.E.M. | Andreastuti S.D., | 2.543 |
| Land subsidence of Jakarta (Indonesia) and its relation with urban development | 2011 | Natural Hazards | 49 | Abidin H.Z., Andreas H., Gumilar I., Fukuda Y., Pohan Y.E., Deguchi T. | Abidin H.Z., Andreas H., Gumilar I., Pohan Y | 1.719 |
| Examination of three practical run-up models for assessing tsunami impact on highly populated areas | 2011 | Natural Hazards and Earth System Science | 27 | Muhari A., Imamura F., Koshimura S., Post J. | Muhari A., | 1.735 |
| Objective rapid delineation of areas at risk from block-and-ash pyroclastic flows and surges | 2009 | Bulletin of Volcanology | 27 | Widiwijayanti C., Voight B., Hidayat D., Schilling S.P. | Widiwijayanti C., Hidayat D., | 2.519 |
| Proposed seismic hazard maps of Sumatra and Java islands and | 2008 | Journal of Earth System Science | 26 | Irsyam M., Dangkua D.T., Hendriyaw | Irsyam M., Dangkua D.T., Hendriyawa | 1.040 |




| Title | Year | Journal Name | No. of citation | Authors | Indonesian author(s) | Impact Factor |
|---|---|---|---|---|---|---|
| during the 2007 earthquake sequence | | | | Sieh K., Fang P., Li Z., Galetzka J., Genrich J., Chlieh M., Natawidjaja D.H., Bock Y., Fielding E.J., Ji C., Helmberger D.V. | | |
| Circum-Pacific seismic potential: 1989-1999 | 1991 | Pure and Applied Geophysics PAGEOPH | 249 | Nishenko S.P. | | 1.618 |
| Middle paleolithic assemblages from the Indian subcontinent before and after the Toba super-eruption | 2007 | Science | 189 | Petraglia M., Korisettar R., Boivin N., Clarkson C., Ditchfield P., Jones S., Koshy J., Lahr M.M., Oppenheimer C., Pyle D., Roberts R., Schwenninger J.-L., Arnold L., White K. | | 33.61 |
| Historical eruptions of Merapi Volcano, Central Java, Indonesia, 1768-1998 | 2000 | Journal of Volcanology and Geothermal Research | 167 | Voight B., Constantine E.K., Siswowidjoyo S., Torley R. | Siswowidjoyo S., | 2.543 |
| Tracking the rupture of the Mw = 9.3 Sumatra earthquake over 1,150 km at teleseismic distance | 2005 | Nature | 161 | Kruger F., Ohrnberger M. | | 41.456 |
| Limited global change due to | 2002 | Quaternary Science | 146 | Oppenheimer C. | | 4.572 |

| Title | Year | Journal Name | No of citations | Authors | Indonesian author(s) | Impact Factor |
|---|---|---|---|---|---|---|
| microzonation study of Jakarta city, Indonesia | | | | an, Hoedajianto D., Hutapea B.M., Kertapati E.K., Boen T., Petersen M.D. | n, Hoedajianto D., Hutapea B.M., Kertapati E.K., Boen T., Petersen M.D. | |
| Kelut Volcano monitoring: hazards, mitigation and changes in water chemistry prior to the 1990 eruption | 1994 | Geochemical Journal | 24 | Badrudin M. | Badrudin M. | 1.505 |
| Modeling study of growth and potential geohazard for LUSI mud volcano: East Java, Indonesia | 2009 | Marine and Petroleum Geology | 22 | Istadi B.P., Pramono G.H., Sumintadireja P., Alam S. | Istadi B.P., Pramono G.H., Sumintadireja P., Alam S. | 2.639 |
| Ground-motion attenuation relationship for the Sumatran megathrust earthquakes | 2010 | Earthquake Engineering and Structural Dynamics | 22 | Megawati K., Pan T.-C. | Megawati K | 2.305 |
| The threat of hazards in the Semeru volcano region in East Java, Indonesia | 1997 | Journal of Asian Earth Sciences | 15 | Siswowidjoyo S., Sudarsono U., Wirakusumah A.D. | Siswowidjoyo S., Sudarsono U., Wirakusumah A.D. | 2.741 |
| 1. | | | | | | |

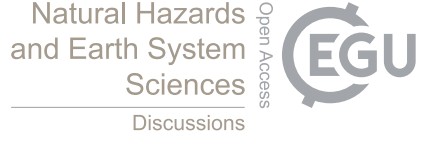

| Title | Year | Journal Name | No. of citation | Authors | Indonesian author(s) | Impact Factor |
|---|---|---|---|---|---|---|
| the largest known Quaternary eruption, Toba ≈74 kyr BP? | | Reviews | | | | |
| | | | Total citations: 2520 | Total number of authors: 65 | Total number of Indonesian authors: 10 | Total IF: 174.557 |

| Title | Year | Journal Name | No of citations | Authors | Indonesian author(s) | Impact Factor |
|---|---|---|---|---|---|---|
| | | | Total citations: 279 | Total number of authors: 35 | Total number of Indonesian authors: 25 | Total IF: 18.746 |





### 3.2.3.3 Topic on disaster risk reduction

This sub-section examines the roles of Indonesian researchers in the publications of materials related to the topic of disaster risk reduction. As can be seen from Figure 25, there are twice the number of authors in the most cited papers than the

Indonesians, while the opposite occurs in when comparing the 10 cited papers with Indonesian as first author. While the combined impact factor does not differ greatly, the total combined impact factor and citations are still 8 times greater when comparing 10 most cited papers in general to those first authored by Indonesians. Table 14 shows the 10 most cited papers in DRR group. Most papers discuss various aspects reviewing the 2004 tsunami recovery and reconstructions, from the building back-better (Kennedy *et al.*, 2008), humanitarian system (Telford and Cosgrave, 2007), institutional (Birkmann *et*

*al.*, 2010) and ethnic groups (Gaillard et al., 2008b) responses, the role of the environment (Srinivas and Nakagawa, 2008), housing (Steinberg, 2007), peace (Le Billon and Waizenegger, 2007). The other papers discuss tsunami warning in Padang (Taubenböck *et al.*, 2009) and disasters in general. This implies that Indonesian authors needed to collaborate more with international authors. Setiadi, formerly from UNU-EHS, was involved in two publications related to the GITEWS program following the 2004 Indian Ocean tsunami. This shows how lacking the quality and impact of publications written by

Indonesian scholars in this topic. Djalante has 3 papers within the list of those first authored by Indonesian, on her publications related to the review of DRR governance in Indonesia (Table 12).

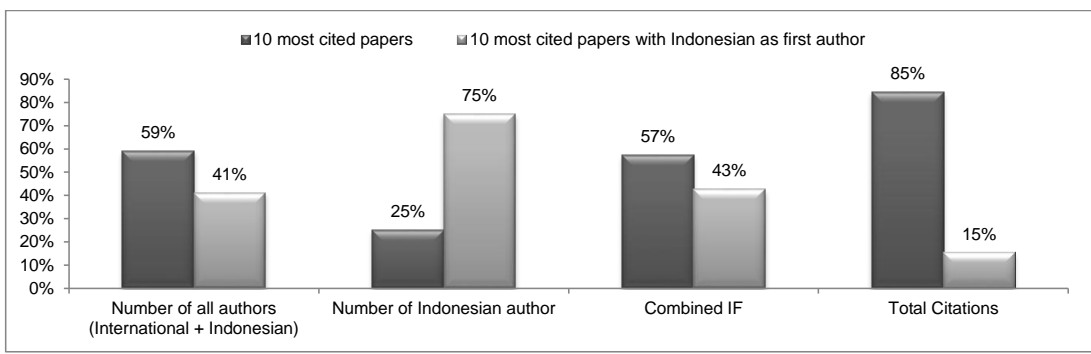

**Figure 25 Comparing the role of Indonesian authors in 10 most cited publications in DRR categories**




**Table 12 List of 10 most cited paper in the disaster risk reduction categories**

| Title | Year | Journal Name | No. of citation | Authors | Involvement of Indonesian author | Impact Factor |
|---|---|---|---|---|---|---|
| Building resilience to natural hazards in Indonesia: Progress and challenges in implementing the Hyogo Framework for Action | 2012 | Natural Hazards | 30 | Djalante R., Thomalla F., Sinapoy M.S., Carnegie M. | Djalante R., Sinapoy M.S., | 1.719 |
| The role of fire in changing land use and livelihoods in Riau-Sumatra | 2004 | Ecology and Society | 29 | Suyanto S., Applegate G., Permana R.P., Khususiyah N., Kurniawan I. | Suyanto S., Permana R.P., Khususiyah N., Kurniawan I | 3.310 |
| Review Article: Adaptive governance and resilience: The role of multi-stakeholder platforms in disaster risk reduction | 2012 | Natural Hazards and Earth System Science | 20 | Djalante R. | Djalante R. | 1.735 |
| Landslide hazard and community-based risk reduction effort in Karanganyar and the surrounding area, central Java, Indonesia | 2011 | Journal of Mountain Science | 18 | Karnawati D., Fathani T.F., Ignatius S., Andayani B., Legono D., Burton P.W. | Karnawati D., Fathani T.F., Ignatius S., Andayani B., Legono D. | 0.963 |
| Coastal flood management in Semarang, Indonesia | 2008 | Environmental Geology | 18 | Marfai M.A., King L. | Marfai M.A | 1.026 |
| Indonesia and the tsunami: Responses and foreign policy implications | 2006 | Australian Journal of International Affairs | 17 | Sukma R. | Sukma R. | 0.247 |

| Title | Year | Journal Name | No. of citation | Authors | Involvement of Indonesian author | Impact Factor |
|---|---|---|---|---|---|---|
| The meaning of 'build back better': Evidence From post-tsunami Aceh and Sri Lanka | 2008 | Journal of Contingencies and Crisis Management | 166 | Kennedy J., Ashmore J., Babister E., Kelman I. | | 0.568 |
| The international humanitarian system and the 2004 Indian Ocean earthquake and tsunami | 2007 | Disasters | 116 | Telford J., Cosgrave J. | | 0.742 |
| Extreme events and disasters: A window of opportunity for change? Analysis of organizational, institutional and political changes, formal and informal responses after mega-disasters | 2010 | Natural Hazards | 102 | Birkmann J., Buckle P., Jaeger J., Pelling M., Setiadi N., Garschagen M., Fernando N., Kropp J. | Setiadi N., (UNU-EHS) | 1.719 |
| Environmental implications for disaster preparedness: Lessons Learnt from the Indian Ocean Tsunami | 2008 | Journal of Environmental Management | 101 | Srinivas H., Nakagawa Y. | | 2.723 |
| Housing reconstruction and rehabilitation in Aceh and Nias, Indonesia-Rebuilding lives | 2007 | Habitat International | 97 | Steinberg F. | | 1.746 |
| Peace in the wake of disaster? Secessionist conflicts and the 2004 Indian Ocean tsunami | 2007 | Transactions of the Institute of British Geograph | 88 | Le Billon P., Waizenegger A. | | 3.636 |


| Title | Year | Journal Name | No. of citation | Authors | Involvement of Indonesian author | Impact Factor |
|---|---|---|---|---|---|---|
| "Last-Mile" preparation for a potential disaster - Interdisciplinary approach towards tsunami early warning and an evacuation information system for the coastal city of Padang, Indonesia | 2009 | Natural Hazards and Earth System Science | 68 | Taubenbock H., Goseberg N., Setiadi N., Lammel G., Moder F., Oczipka M., Klupfel H., Wahl R., Schlurmann T., Strunz G., Birkmann J., Nagel K., Siegert F., Lehmann F., Dech S., Gress A., Klein R. | Setiadi N (UNU-EHS) | 1.735 |
| Ethnic groups' response to the 26 December 2004 earthquake and tsunami in Aceh, Indonesia | 2008 | Natural Hazards | 67 | Gaillard J.-C., Clave E., Vibert O., Azhari D., Denain J.-C., Efendi Y., Grancher D., Liamzon C.C., Sari D.R., Setiawan R. | Azhari D., Efendi Y., Sari D.R., Setiawan R | 1.719 |
| | | | Total citations: 958 | Total number of authors: 49 | Total number of Indonesian authors: 7 | Total IF: 18.610 |

| Title | Year | Journal Name | No. of citation | Authors | Involvement of Indonesian author | Impact Factor |
|---|---|---|---|---|---|---|
| Bridging the gaps: The role of local government capability and the management of a natural disaster in Bantul, Indonesia | 2012 | Natural Hazards | 14 | Kusumasari B., Alam Q. | Kusumasari B., | 1.719 |
| Tsunami mitigation efforts with pTA in west Sumatra province, Indonesia | 2010 | Journal of Earthquake and Tsunami | 10 | Muhari A., Imamura F., Natawidjaja D.H., Diposaptono S., Latief H., Ismail F.A. | Muhari A., D.H., Diposaptono S., Latief H., Ismail F.A. | 0.431 |
| Pathways for adaptive and integrated disaster resilience | 2013 | Natural Hazards | 10 | Djalante R., Holley C., Thomalla F., Carnegie M. | Djalante R., | 1.719 |
| Influence of the institutional and socio-economic context for responding to disasters: Case study of the 1994 and 2006 eruptions of the Merapi Volcano, Indonesia | 2012 | Geological Society Special Publication | 8 | Mei E.T.W., Lavigne F. | Mei E.T.W., | 2.580 |
| | | | Total citations: 174 | Total number of authors: 34 | Total number of Indonesian authors: 21 | Total IF: 13.808 |



### 3.2.3.4 Topics on climate change


This sub-section examines the roles of Indonesian researchers in the publications of materials related to the topic of climate change impacts and governance. As can be seen from Figure 26, there are more collaborations take place amongst authors in 10 most cited papers while almost 80 percent authors of 10 most cited papers with Indonesian as first authors, are Indonesian. While the impact factor of the papers differ greatly (9 times), the citations by the Indonesian first authors' publications catch up. Table 13 shows that majority of the papers talk about the Indonesian forest fires in relation to climate change. The other papers talk about observations of changes in rainfall, drought and temperature. Page et al has their paper on the amount of carbon released from forest fires as the highest cited paper in this category. There are two Indonesians, Jaya A., Limin S from Palangkaraya University in Kalimantan are involved in the most cited paper in this group on their paper (Page et al., 2002). Aldrin and have also published widely cited papers related to observations on changes in temperature and rainfall in Indonesia (Aldrian and Djamil, 2008; Aldrian and Dwi Susanto, 2003). Amien and Redjekiningrum from the Center for Soil and Agroclimate Research have collaborated in two papers on the examinations of possible climate change on rice production in Java (Amien *et al.*, 1999). Marfai from UGM has two papers that examine the impacts of sea level rise on the coastal areas in Semarang (Marfai and King, 2008b; Marfai et al., 2008).



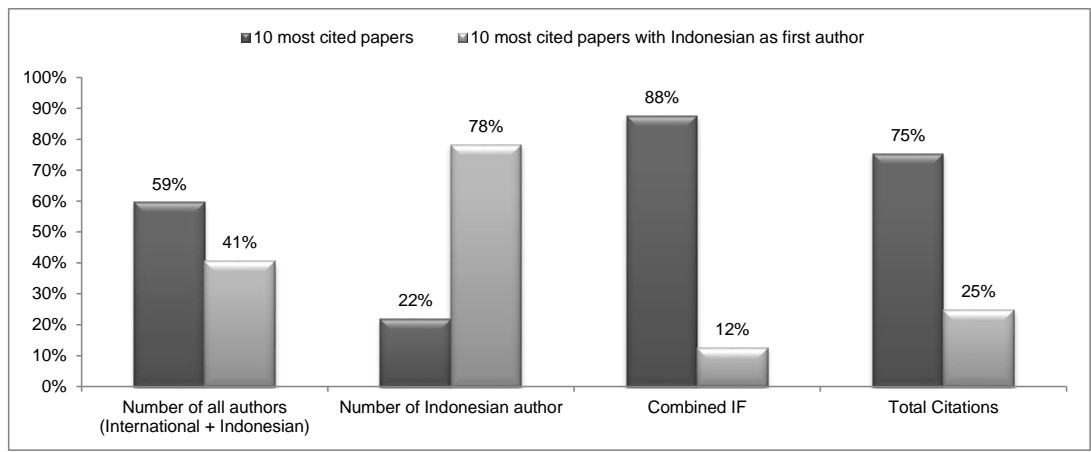

**Figure 26 Comparing the role of Indonesian authors in 10 most cited publications**


**Table 13 List of 10 most cited papers in the topic of climate change**

| Title | Year | Journal Name | No of citation | authors | Indonesian author | Impact Factor |
|---|---|---|---|---|---|---|
| The amount of carbon released from peat and forest fires in Indonesia during 1997 | 2002 | Nature | 633 | Page S.E., Siegert F., Rieley J.O., Boehm H.-D.V., Jaya A., Limin S. | Jaya A., Limin S. | 41.456 |
| Increased damage from fires in logged forests during droughts caused by El Niño | 2001 | Nature | 309 | Siegert F., Ruecker G., Hinrichs A., Hoffmann A.A. |  | 41.456 |
| Identification of three dominant rainfall regions within Indonesia and their relationship to sea surface temperature | 2003 | International Journal of Climatology | 114 | Aldrian E., Dwi Susanto R. | Aldrian E., Dwi Susanto R. | 3.157 |
| Climate regulation of fire emissions and deforestation in equatorial Asia | 2008 | Proceedings of the National Academy of Sciences of the United States of America | 192 | Van Der Werf G.R., Dempewolf J., Trigg S.N., Randerson J.T., Kasibhatla P.S., Giglio L., Murdiyarso D., Peters W., Morton D.C., Collatz G.J., Dolman A.J., DeFries R.S. | Murdiyarso D., | 9.674 |
| Identification of three dominant rainfall regions within Indonesia and their relationship to sea surface temperature | 2003 | International Journal of Climatology | 282 | Aldrian E., Dwi Susanto R. | Aldrian E., Dwi Susanto R. | 3.157 |
| Upwelling along the coasts of Java and Sumatra and its relation to ENSO | 2001 | Geophysical Research Letters | 137 | Susanto R.D., Gordon A.L., Zheng Q. | Susanto R.D. | 4.196 |
| Effects of interannual climate variability and climate change on rice yield in Java, Indonesia | 1996 | Water, Air, and Soil Pollution | 46 | Amien I., Rejekiningrum P., Pramudia A., Susanti E. | Amien I., Rejekiningrum P., Pramudia A., Susanti E. | 1.554 |
| Simulated rice yields as affected by interannual climate variability and possible climate change in Java | 1999 | Climate Research | 14 | Amien I., Redjekiningrum P., Kartiwa B., Estiningtyas W. | Amien I., Redjekiningrum P., Kartiwa B., Estiningtyas W. | 2.496 |

Natural Hazards
and Earth System
| Title | Year | Journal Name | No of citation | authors | Indonesian author | Impact Factor |
|---|---|---|---|---|---|---|
| Human amplification of drought-induced biomass burning in Indonesia since 1960 | 2009 | Nature Geoscience | 98 | Field R.D., Van Der Werf G.R., Shen S.S.P. | | 11.740 |
| Upwelling along the coasts of Java and Sumatra and its relation to ENSO | 2001 | Geophysical Research Letters | 81 | Susanto R.D., Gordon A.L., Zheng Q. | Susanto R.D | 4.196 |
| Locating REDD: A global survey and analysis of REDD readiness and demonstration activities | 2011 | Environmental Science and Policy | 69 | Cerbu G.A., Swallow B.M., Thompson D.Y. | | 3.018 |
| Monsoon drought over Java, Indonesia, during the past two centuries | 2006 | Geophysical Research Letters | 52 | D'Arrigo R., Wilson R., Palmer J., Krusic P., Curtis A., Sakulich J., Bijaksana S., Zulaikah S., Ngkoimani L.O. | Bijaksana S., Zulaikah S., Ngkoimani L.O. | 4.196 |
| Using NOAA/AVHRR products to monitor El Niño impacts: Focus on Indonesia in 1997-98 | 2000 | Bulletin of the American Meteorological Society | 34 | Gutman G., Csiszar I., Romanov P. | | 11.810 |

| Title | Year | Journal Name | No of citation | authors | Indonesian author | Impact Factor |
|---|---|---|---|---|---|---|
| Spatio-temporal climatic change of rainfall in East Java Indonesia | 2008 | International Journal of Climatology | 13 | Aldrian E., Djamil Y.S. | Aldrian E., Djamil Y.S. | 3.157 |
| Potential climate-change related vulnerabilities in Jakarta: Challenges and current status | 2011 | Habitat International | 10 | Firman T., Surbakti I.M., Idroes I.C., Simarmata H.A. | Firman T., Surbakti I.M., Idroes I.C., Simarmata H.A. | 1.577 |
| The impact of tidal flooding on a coastal community in Semarang, Indonesia | 2008 | Environmentalist | 8 | Marfai M.A., King L., Sartohadi J., Sudrajat S., Budiani S.R., Yulianto F. | Marfai M.A., Sartohadi J., Sudrajat S., Budiani S.R., Yulianto F. | 0.0 |
| REDD+ and Forest Governance in Indonesia: A Multistakeholder Study of Perceived Challenges and Opportunities | 2013 | Journal of Environment and Development | 7 | Mulyani M., Jepson P. | Mulyani M | 1.824 |
| Potential vulnerability implications of coastal inundation due to sea level rise for the coastal zone of Semarang city, Indonesia | 2008 | Environmental Geology | 6 | Marfai M.A., King L. | Marfai M.A., | 1.078 |

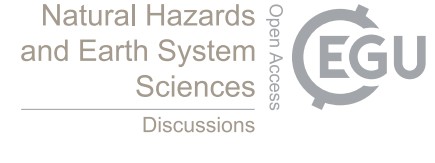

| Title | Year | Journal Name | No of citation | authors | Indonesian author | Impact Factor |
|---|---|---|---|---|---|---|
| Large aerosol radiative forcing due to the 1997 Indonesian forest fire | 2003 | Geophysical Research Letters | 29 | Podgorny I.A., Li F., Ramanathan V. | | 4.196 |
| | | | Total citations: 1611 | Total number of authors: 47 | Total number of Indonesian authors: 7 | Total IF: 134.899 |

| Title | Year | Journal Name | No of citation | authors | Indonesian author | Impact Factor |
|---|---|---|---|---|---|---|
| Governing carbon, transforming forest politics: A case study of Indonesia's REDD+ Task Force | 2015 | Asia Pacific Viewpoint | 4 | Astuti R., Mcgregor A. | Astuti R., | 0.860 |
| | | | Total citations: 527 | Total number of authors: 32 | Total number of Indonesian authors: 25 | Total IF: 19.039 |



### 4 Conclusions and recommendations for future research needs

This paper has outlined an overview of current research trends and progress related to hazards, disasters, and disaster risks reduction, as well as increasingly on climate change impacts and governance in Indonesia. The first recommendation is that future research agendas also need to focus on different hazards, different locations in Indonesia, and other topics in DRR and climate change. It has been shown in this paper that the research have focused mainly on the geophysical hazards and those related to hydro-meteorological hazards only receive attention recently. Assessments of multi- hazards that combined risks and the associated impacts from geophysical and hydro-meteorological hazards simultaneously are suggested.

It has been seen that majority of research focus on the Islands of Java and Sumatera. There is still greater need for research on climate change topics related to vulnerability, adaptation limits, loss and damage, impacts on key sectors such as fisheries, coastal communities, food security and health. There needs to be more research on other part of Indonesian. The impacts of sea level rise on small islands, drought on forest in Kalimantan and Papua, increase sea water and ocean acidification on fisheries industry in Sulawesi and eastern part of Indonesia, are some of the increasingly worrisome expected from climate change. More research is needed on the context of urban areas by which social risks and risks from natural hazards play out simultaneously, and the impacts on the urban dwellers are to be understood. The governance of DRR has not received many researches especially on the interplay with decentralization which put responsibility for disaster risk management and reduction at the local government level. Many activities done by international and development agencies have focused on the community level. There is abundance of activities reports by donor and international agencies, however, those reports rarely be made available or submitted for academic publications.

The second recommendation is on the need to strengthen the capacity of research collaborations between Indonesian and international researchers, multi-disciplinarity of research and publications for high impacts journals. It is clear that some of the very limited Indonesian researchers from ITB, LIPI, and UGM have been involved in international collaborations and in publications of high impacts journal. There is abundance of materials within Indonesian repositories related to *bencana* (disaster in English), especially within the repositories with ITB, UGM, and UNSYIAH. These materials and research activities done within the universities needed to be reviewed and submitted for international journals in order to give a broader view on issues that have been discussed by scholars in Indonesia. There is only 1 Indonesian journal that made to the list.

There is a need for better target of scholars to do more collaboration for research and writing for high impact journals. It is also not clear how collaborations amongst scientists from social and physical scientist have taken place in Indonesia. There is still small numbers of female and of early career researchers. Moreover, the roles of universities and researchers from outside Java had been very limited in their progress. There is increasing call for a more inter-disciplinarily collaborations so that complex problems on the social and environmental issues can be understood better and problems identifications can target those in needs better. It is also not clear how or whether science and policy collaborations have taken place and be documented in these listed publications. Although we can see from the list that some of the most prominent authors are not only from universities but also from national level government agencies. The roles of private business and the communities at risk have rarely been part of the research and collaborations.

In conclusion this study has been able to determine the progress in research related to hazards, risks, and risk deduction and climate change in Indonesia. It has also been able to examine the roles of Indonesian scientist in collaborations and towards high quality publications. The recommendations are outlined toward these two issues and it is the responsibility both by the





Indonesian and international organizations that have and going to work in Indonesia to be able to meet the needs in order for

Indonesia to better understood and manage its hazards and risks in the future.

**Appendix**

**Appendix 1 List of Indonesian authors with publications more than 1**

| Indonesian Author | No. of public­ations in this review | Org. | Location | | | | Gender | | SCOPUS profile (publications, citations, h-index, co-authors, most frequent collaborator) | Google Scholar profile (publications, citations, h-index,i-10 index) |
|---|---|---|---|---|---|---|---|---|---|---|
| | | | J | OJ | N | I | M | F | | |
| Surono, | 18 | PVMBG | | | x | | x | | 27, 348, 12, 125, Hendrasto M | NA |
| Abidin, H.Z. | 16 | ITB | X | | | | x | | 53, 493, 11, 121, Andreas H | NA |
| Natawidjaja, D.H. | 11 | LIPI | | | x | | x | | 42, 1913, 21, 123, Sieh KE | 147, 2964, 25, 33 |
| Marfai, M.A. | 11 | UGM | x | | | | x | | 19, 183, 8, 36, King, Lorenz | 79, 517, 12, 14 |
| Hendrasto M | 10 | PVMBG | | | x | | x | | 16, 92, 4, Surono | NA |
| Andreas, H. | 10 | ITB | x | | | | x | | 20, 123, 6, 46, Abidin, H Z | NA |
| Ratdomopurbo, A. | 8 | NTU | | | | x | x | | 17, 441, 10, 59, Lühr, B G | NA |
| Muhari, A. | 8 | MAAF | | | x | | x | | 15, 112, 6, 53, Imamura, F | NA |
| Sumarti, Sri. | 8 | UGM | x | | | | | x | 14, 367, 13, 84, Surono | NA |
| Suwargadi, BW | 7 | LIPI | | | x | | x | | 31, 1102, 17, 103, Natawidjaja, DH | 97, 1585, 20, 24 |
| Meilano, Irwan | 7 | ITB | x | | | | x | | 32, 249, 8, 115, Fumiaki | 95, 357, 9, 9 |
| Setiadi, N. | 7 | Formerly UNU-EHS, Germany | | | | x | | x | 9, 105, 4,4, 47, Birkmann, J | NA |
| Djalante, R. | 6 | UNU-EHS, Germany, | | | | x | | x | 7, 56, 4, 40, Thomalla, F | 20, 146, 6, 6 |
| Andreastuti, S | 6 | PVBMG | | | x | | | x | 6, 256, 6, 39, del Marmol, M A | NA |
| Bronto, S. | 6 | PVBMG | | | x | x | | | | NA |
| Purbawinata M. A. | 6 | PVBMG | | | x | | x | | 8; 114; 5; 56; Hendrasto, M | NA |
| Fathani, T.F. | 5 | UGM | x | | | | x | | 11;9;1;26;Karnawati D | NA |
| Kirono, D.G.C. | 5 | CSIRO (Australia) | | | | x | | x | 25,399,9,65,Kent DM | NA |
| Gumilar I | 5 | ITB | x | | | | x | | 12,46,3,28,Zainal HA | NA |
| Gamal, M | 5 | ITB | x | | | | x | | 11,41,3,37, Zainal HA | NA |
| Hadmoko, D.S. | 5 | UGM | x | | | | x | | 11,84,5,55,Lavigne F | NA |
| Harjono, H. | 5 | LIPI | | | x | | x | | 13,143,8,40,Diament M | NA |
| Prasetya, G. | 5 | TRC | | | | x | x | | 17, 230, 7, 57, De Lange, W P | NA |
| Syamsidik, | 5 | UNSYIAH | x | | | | x | | 8,8,1,15,Aoki S | 23; 32; 4; 1 |
| Habibi, P | 4 | UNRAM | x | | | | x | | 4,9,1,27,Butler JR | NA |
| Hananto, N.D. | 4 | LIPI | | | x | | x | | 13,354,9,56,Singh S | 53;520;10;10 |
| Handayani T | 4 | UNRAM | x | | | | | x | 4,9,1,27, Butler JR | NA |
| Latief, H. | 4 | ITB | x | | | | x | | 1,0,0,5,Asrurifak M | NA |
| Triatmadja, R. | 4 | UGM | x | | | | x | | 5,6,1,5,Benazir | 57; 50;4;2 |
| Irsyam, M. | 4 | ITB | x | | | | x | | 20,63,4,75,Hryciw RD | NA |





| | | | | | | | |
|---|---|---|---|---|---|---|---|
| **Kongko, W.** | 3 | BPPT | | x | x | 12,269,5,69,McAdoo BG | 35;422;7;5 |
| **Subandriyo D A** | 3 | BPPTKG | | x | x | 2,15,1,17,Agung Nandaka IM | NA |
| **Harjadi, P.** | 3 | BMKG | | x | x | 10,138,7,45,Yamashina T | NA |
| **Purnomo, H.** | 3 | IPB | x | | x | 19,151,7,31,Irawati RH | 377;563;12;14 |
| **Yulianto, F.** | 3 | LAPAN | | x | x | 6,38,3,17,Komarudin MR | NA |
| **Karnawati, D.** | 3 | UGM | x | | x | 11;36;3;26;Fathani, T F | NA |
| **Pribadi, K.S.** | 3 | ITB | x | | x | 9,12,2,26,Soekiman/Sumardi,Wirahadikusumah | NA |
| **Djaja R** | 3 | Bakosurtanal | | x | x | 3,95,3,14,Abidin HZ | NA |
| **Mulyasari, F.** | 3 | ITB | x | | x | 7,10,2,Shaw R | NA |
| **Suadnya W** | 3 | UNRAM | x | | x | 5,9,1,33, Butler JR | NA |
| **Yanuartati, Y** | 3 | UNRAM | x | | x | 3,9,1,23,Bohensky EL | NA |
| **Darmawan D** | 3 | Telkom Uni | x | | x | 6,92,3,23,Abidin HZ | NA |
| **Murdiyarso, D.** | 3 | IPB | x | | x | 57,2797,23,>150,Verchot LV | 308;6857;39;72 |
| **Susanto, R.D.** | 2 | University Maryland (USA) | | x | x | 23,837,15,38,Gordon AL | 55;1826;20;26 |
| **Firman, T** | 2 | ITB | x | | x | 8,21,2,5,Hudalah D | 70;1264;23;28 |
| **Sagala, S** | 2 | ITB | x | | x | 5,25,2,12,Okada N | 17 publications 34 citations 1.45 impact points |
| **Lassa, JA** | 2 | NTU | | x | x | 6,0,0,9,Caballero-A M | https://www.rsis.edu.sg/profile/jonatan-anderias-lassa/#.VsrWBPnhC70 |
| **Andayani, B.** | 2 | UGM | x | | x | 5,8,1,9,Fathani TF | 30;91;6;2 |
| **Siswowidjoyo S.,** | 2 | PVMBG | | x | x | 3,170,3,7,Constantine EK | NA |
| **Aldrian, E** | 2 | BMKG | | x | x | 13,268,8,46,Podzun R | 62;1099;11;15 |
| **Prawirodirdjo, L** | | UC San Diego | | x | x | 18,1799,16,59,Bock Y | NA |
| **Amien, I** | 2 | IPB | x | | x | 2,36,6,Estinintyas W | NA |
| **Redjekiningrum, P** | 2 | IPB | X | | x | 1,14,1,3,Amien I | NA |
| **Sudibyakto** | 2 | UGM | x | | x | 3,0,0,2,Abasi/Haroonah | NA |

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
