# Peer review of "Research trends on hazards, disasters, risk reduction and climate change in Indonesia: a systematic literature review"

_Natural Hazards and Earth System Sciences, 2016_

## Short Comment (SC1) · 25 May 2016

The discussion paper has given a good overview on the development of the research on disaster risk and climate change related issues in Indonesia. I would like to comment on the following issues

With regard to the overall topics and place where most reseach have been identified: It has been demostrated in the paper that the publication on the DRR / DRM has the lowest citation rate, indicating a possible slower take-up of the publications and/or that the topics of "hazads, risks and disasters" has been more of the most interest of the research communities. It is necessary to support more research activities in this field, quantitatively and qualitatively, since DRR is, to my opinion, an important and

necessary follow-up of the knowledge about "hazards, risks, and disasters". There have been – and should be more – efforts to reduce disaster risk in Indonesia that need to be scrutinize scientifically and continuously impoved also by means of science. Furthermore the finding of this paper outlines that the publication in this topic tends to decrease, which may be or may not be linked with the "trend" of research and political activities focusing on climate change. Indonesia has been dealing and will continue to deal with various hazards and I would like to maintain the importance of geophysical hazards and other hazards which are not directly linked with climate change.

I also observed the tendency that researches have been focusing on Java and Sumatra, and agree with the author that more researchs in other areas are needed, especially in the eastern part of Indonesia. Additionally, it may be helpful to locate areas where the research activities and measures are needed by existing hazard (and vulnerability) maps (e.g. from BNPB or existing research findings) to be able to focus on the current issues.

Other issue related to the citation number and impact factor, high citation number of papers that contain mega-disasters (e.g. Indian Ocean tsunami) might have more citations may be explained by more international attention on those events. It might support the citation/impact rate if the publication is linked to such events and current international issues, however this does not downplay the importance of research on small-scale but frequent as well as slow-onset hazard events.

With regard to the role of Indonesian researchers: I agree that most likely there are many other publications, which are not submitted as peer-reviewed articles in the well-known international journals, conferences, or books, which were not counted here and might provide different figures that the finding in the paper. Those may be research reports or papers in national publications e.g. LIPI publication series, ITB Journals etc., or relevant PhD theses. Indeed I would also see that the finding in the paper highlights the – still – lack of publication culture and limited access to publishing in international journals in Indonesia, despite of many research activities by Indonesian researchers,

especially the case when there is no collaboration with international researchers / institutions. Capacity building of Indonesian researchers to meet international publication standards is still necessary. Additionally, language and writing skills need to be improved. Collaboration with international researchers / research institutions may foster improvement of publication skills, but also involvement in international events will be a good opportunity to get up-to-date on the global issues or publication topics which is "in", as well as to make own research activities known among the international "peers" doing research in the disaster-related fields. I have met and believe in many young Indonesian scientists, highly motivated and bright ones, which have the potential to alter the publication quantity and quality in international arena. Collaboration and networking between Indonesian scholars who have studied abroad to do joint publications may also be helpful.

---

## Author Comment (AC1) · 27 May 2016

Dear Dr Setiady,

Many thanks for these extremely constructive comments. I have summarized those comments and mostly turn them into a set of recommendations to be included in the last section on recommendations for future research.

1.There need to review existing publications which are not listed in well-known international journals, or conferences or books. Should it be included here, the findings might be different. a.Reply: many thanks for this comment and I will put this as disclaimer statement or factors for consideration in the method section

2.Identification of further research agendas on: a.DRR, qualitatively and quantitatively, as the follow of the abundance of knowledge on HRD b.geophysical hazards and other hazards which are not directly linked with climate change c.areas beyond Sumatra and Java d.utilizing available knowledge on hazards maps already available from BNPB or others e.small-scale but frequent and slow-onset hazards events f.the role of science to support DRR in Indonesia g. consultation of research reports and papers in national publications such as LIPI or ITB or theses to identify research agendas within Indonesian scholars

3.Capacity building for Indonesian researchers through: a.Still lack of publication culture and limited access to publishing in International journals, despite of many research activities by Indonesian researchers, especially then there is no collaboration with international researchers / institutions b.Capacity building of Indonesian researches to meet international publications standards is still necessary c.Language and writing skills to be improved d.Collaboration with international researchers / research institutions may foster improvement in publications skills, but also involvement in international events will be a good opportunity to keep up-to-date on the global issues or publications topics with is researched, as well as to make own research activities known among the international 'peers 'doing research in the disaster-related field e.Collaborations and networking between Indonesian scholars who have study abroad to do join publications is also needed.

---

## Referee Comment (RC1) · Anonymous Referee #1 · 8 Jun 2016

The paper presents a systematic literature review of the research trends on natural hazards, DRR and climate change in Indonesia. This paper has the potential to be very impacting in the scientific community enhancing new relevant topics to be further investigated in the Indonesian country. However there is the need of a hard reworking for several reasons, for which I suggest a major revision. However, I think that if all the comments suggested will be not addressed adequately the paper would be not suitable for publishing. In addition an English language revision is needed.

1-The abstract needs to be summarized, avoiding useless details (for ex. line 24-25, the number of publications per stage etc). I suggest writing no more than 350 words.

2-The introduction lacks of research gaps identification. The author should emphasize

the importance of this literature review (e.g., Sendai Framework for DRR) by developing a more solid introduction that would bring the reader to the following chapters. It lacks comments on the topics involved in the review. For instance the author needs to clarify in which context "climate change" has been considered (i.e. broad sense or related to natural disasters). The same should be done for all the themes (disasters and risk and DRR). In addition, I think it is not necessary to list more than 20 review papers (lines 85-91) showing the same methodology based on a whole range of different topics.

3-In the methodology there is no clear delineation of the timeline the author chose for selecting the paper for the reviewing process.

4-The results section lacks of comments, trends and justifications of the results obtained. There is the need to elaborate the findings and give some interpretations to them without being repetitive. The decision to develop the two objectives separately is good. However the many sub-chapters created made the paper redundant (in term of results and charts presented) and difficult to read. This is valid for both the objectives. A general rearranging of the structure of the paper is needed.

A review paper is a useful tool to give other researchers the state of the art of the current research and advances. It is not just a mere list of the topics of the papers found. As it is impossible to mention all the papers ($\approx$750 are too many) the author needs to justify the methodology of citation (the most recent, the most important, the most cited etc) and provide added comments.

Regarding the second objective (i.e. authorship) there are too many abbreviations that need to be expressed at least once and an additional explanation is needed for the provided tables. Moreover, at line 409 the author considered the gender of the authorship of the selected papers. I think this should need more emphasis, consideration and background.

5-There is a general lack of discussion in the results section that determines the poor conclusions and recommendations for further research. There is just a mention of the

tips for further research that need to be enriched.

6-Appendix 1 gives no added value to the paper.

7-Figures and tables: (a) There are too many tables and figures that do not give any additional value to the review. Most of them can be easily replaced with one or two sentences in the text. (b) In many of the bar charts the sum the author provided in the caption does not match the real sum showed by the bars. This bias has been found in some figures and tables. Is this a lack of attention or a justifiable bias? In addition, in Table 5, the citation average of the first row should be 8.21 not 8.0. Please check all of the figures, tables and captions. (c) Generally the captions lacks totally of details, are poor in content and sometimes of agreement. There are no references of the timeframe, places etc, and some of the charts lack of x or y labels. (d) The hazard map (Fig.1) presented in the introduction lacks of a legend expressing the colors (supposed to show the level of hazard) and the reference is missing in the reference list. I think that a risk map would be suitable to show the risk profile of the country since Risk is defined by Hazard x Vulnerability x Exposure.

---

## Author Comment (AC2) · 29 Jun 2016

GENERAL COMMENTS: •The paper presents a systematic literature review of the research trends on natural hazards, DRR and climate change in Indonesia. •This paper has the potential to be very impacting in the scientific community enhancing new relevant topics to be further investigated in the Indonesian country. •However there is the need of a hard reworking for several reasons, for which I suggest a major revision. •However, I think that if all the comments suggested will be not addressed adequately the paper would be not suitable for publishing. •In addition an English language revision is needed. AUTHOR'S REPLY: The author wishes to thanks the

reviewer for the very valuable comments. I will endeavor to revise the article as per general and specific comments. The author has relooked the paper and considerably rewrite and remove one third of the structure since it has brought repetition and confusions to the reviewer.

SPECIFIC COMMENTS: In this section, the author will give the reply immediately after individual number.

1.The abstract needs to be summarized, avoiding useless details (for ex. line 24-25, the number of publications per stage etc). I suggest writing no more than 350 words. Reply: The author has rewritten the abstract and it is now 344 words. More concise summary is given without too much detail.

2.The introduction lacks of research gaps identification. • The author should emphasize the importance of this literature review (e.g., Sendai Framework for DRR) by developing a more solid introduction that would bring the reader to the following chapters. Reply: The author has added two paragraphs explaining the relevance of the study as follow (line 55 – 94): There are two research questions adopted. First is on progress of research on hazards, risks, disasters and climate change in Indonesia, in terms of content and quality. The importances of conducting literature on these topics are several folds. First, the Sendai Framework for DRR has just been adopted and there are extended scope of hazards and risk reduction strategies adopted. The SF-DRR now calls for inclusion of hazards from biological and technological on top of the common natural hazards from geophysical and hydro-climatological hazards (REF). This review will enable identification of hazards that have been the focus of research and those that do not yet receive examination. Second, there is a move from integrated approach to DRR which calls strategies and actions to reduce risks and impacts of those risks, as well as the role of multi actors for DRR. This review will enable identification of strategies that have been undertaken for DRR and hence able to suggest strategies for future DRR and to implement the SFDRR. Third, there is an increasing focus on the impacts of climate change into changing profile of hazards and disasters,

and hence the calls for integrated DRR and CCA to manage climate risks, This review will try to capture whether consideration of climate change risks have been considered as part of research progress in Indonesia. Hence in this paper, the topics considered are grouped into 3 major ones of those on (1) hazards, risks and disasters, (2) disaster risk reduction, and (3) climate change related hazards, vulnerabilities, and risks.

The second research question is related to the roles of Indonesian authors in contributing for research, international publications and collaborations. Determining the progress of Indonesian scholars is important and relevant for several reasons. First, these scholars have most likely lived in Indonesia for considerable amount of time. They have experienced and assessed and examined those social and environmental changes that caused natural hazards and disasters in the first place. These experiences will help them to be more focused and sharp in terms of documenting. Moreover, there have been increasing calls for more case studies especially in the process of IPCC report writing which describe examples from local level. Also, in Indonesia, there is increasing pressure for scholars to write for international journal publications and collaborate. Any outputs from these publications and collaborations are used toward counting their ranks as academics in universities and research institutions (REF). There is also increasing number of Indonesian scholars who have studies abroad mostly in English-speaking education system and have written considerably on various topics related to disasters, risks reduction and climate change impacts (REF). Hence identification of this progress through this systematic review will enable us to determine recent progress undertaken mostly by Indonesian researchers, and hence, can help outlining recommendations for further actions in the future to increase the quality and roles in international spheres.

•It lacks comments on the topics involved in the review. For instance the author needs to clarify in which context "climate change" has been considered (i.e. broad sense or related to natural disasters). Reply: The author added this sentence (line 71-73) Hence in this paper, the topics considered are grouped into 3 major ones of those

on (1) hazards, risks and disasters, (2) disaster risk reduction, and (3) climate change related hazards, vulnerabilities, and risks.

The author added this sentence (line 388-394) The research on climate change is interpreted broadly in this paper. The author include all materials that discuss on impacts of climate change not only on disasters caused by natural hazards but also those in different sectors such as agriculture, forestry, water and health. This is done since the current Sendai Framework for Action calls for multi-risks perspectives (UNISDR, 2015) .

•The same should be done for all the themes (disasters and risk and DRR). Reply: The explanations for these themes has been provided in Table 3 (line 263)

Table 3 Classifications of findings based on topics of research Major topics groups Definitions (UNISDR, 2009) (1) hazard, risks, disasters assessments (HRD)  Hazards: A dangerous phenomenon, substance, human activity or condition that may cause loss of life, injury or other health impacts, property damage, loss of livelihoods and services, social and economic disruption, or environmental damage.  Risks: The combination of the probability of an event and its negative consequences.  Disaster: A serious disruption of the functioning of a community or a society involving widespread human, material, economic or environmental losses and impacts, which exceeds the ability of the affected community or society to cope using its own resources. (2) disaster risk management or reduction (DRR)  The systematic process of using administrative directives, organizations, and operational skills and capacities to implement strategies, policies and improved coping capacities in order to lessen the adverse impacts of hazards and the possibility of disaster (UNISDR).  The concept and practice of reducing disaster risks through systematic efforts to analyze and manage the causal factors of disasters, including through reduced exposure to hazards, lessened vulnerability of people and property, wise management of land and the environment, and improved preparedness for adverse events. (3) climate change vulnerability, impacts and adaptation (CC)  A change of climate which is attributed directly or indirectly to

human activity that alters the composition of the global atmosphere and which is in addition to natural climate variability observed over comparable time periods (UNFCCC).  The adjustment in natural or human systems in response to actual or expected climatic stimuli or their effects, which moderates harm or exploits beneficial opportunities (UNISDR).

Line 264, new sentence In this study, the hazards, risks and disasters are caused by hydro and hydro-climato-meteorological ones (see Table 4).

Line 313, new sentence In this study, DRR include those strategies that are aimed to reduce disaster risks which range from risk management, risk reduction and disaster preparedness activities. The definition is listed in Table 3 previously.

Line 370, new sentence The research on climate change is interpreted broadly in this paper. The author include all materials that discuss on impacts of climate change not only on disasters caused by natural hazards but also those in different sectors such as agriculture, forestry, water and health. This is done since the current Sendai Framework for Action calls for multi-risks perspectives (UNISDR, 2015) .

•In addition, I think it is not necessary to list more than 20 review papers (lines 85-91) showing the same methodology based on a whole range of different topics. Reply: The whole sentences are rewritten (line 103-105) Based on their extensive review on climate change literature, Berrang-Ford et al (Berrang-Ford et al., 2011; 2015) suggested an analytical approach for systematic review and research synthesis as presented in Table 2, which is adopted in this paper.

3.In the methodology there is no clear delineation of the timeline the author chose for selecting the paper for the reviewing process. Reply: A sentence on timeline (1900-2016) is added (in abstract and in method sections) (line 111-112) The author conducts a multi-layered literature review to study publications using the Scopus research engine, with the timeline from 1900 to 2016.

4.The results section lacks of comments, trends and justifications of the results obtained. Reply: The author focuses revision on the structure and adds more discussions on the comments, trends and justifications of the results, so that critical analysis on the progress of research in these topics could be presented.

•There is the need to elaborate the findings and give some interpretations to them without being repetitive. Reply: The author has completely rewritten the conclusion section (line 599-715)

This paper has outlined an overview of current research trends and progress related to hazards, disasters, and disaster risks reduction, as well as increasingly on climate change impacts and governance in Indonesia.

The first recommendation is that future research agendas need to focus on different hazards, different locations in Indonesia, and other topics in DRR and climate change. It has been shown in this paper that the research have focused mainly on the geophysical hazards and those related to hydro-meteorological hazards only receive attention recently. Assessments of multi- hazards that combined risks and the associated impacts from geophysical and hydro-meteorological hazards simultaneously are suggested. It has been seen that majority of research focus on the Islands of Java and Sumatera. This is expected since both islands are the most at risks from natural disasters in Indonesia. However, other islands in Kalimantan, Sulawesi, Maluku and Papua in eastern part of Indonesia have also been impacted by droughts, floods or strong winds. This is needed to be addressed in the future. The impacts of sea level rise on small islands, drought on forest in Kalimantan and Papua, increase sea water and ocean acidification on fisheries industry in Sulawesi and eastern part of Indonesia, are some of the increasingly worrisome expected from climate change.

More research is needed on the context of urban areas by which social risks and risks from natural hazards play out simultaneously, and the impacts on the urban dwellers are to be understood. As world is increasingly urbanized, there is strong attention on

focusing and reducing risks in urban areas through concerted action in a New Urban Agenda from the HABITAT III (UN HABITAT, 2016). Cities in Indonesia like Jakarta, Surabaya or Makassar are rapidly urbanizing and environmental and economic pressures increasing risks the their inhabitants (Santosa, 2000; Firman et al., 2011; Larson et al., 2013; Firman, 2016).

The governance of DRR has not received many researches especially on the interplay with decentralization which put responsibility for disaster risk management and reduction at the local government level. Many activities done by international and development agencies have focused on the community level. There is abundance of activities reports by donor and international agencies on their implementations for DRR or CCA programmes (e.g. USAID Indonesia, 2011; 2015; USAID, 2016), however, those reports rarely be made available or submitted for academic publications. There is still greater need for research on climate change topics related to linkages between poverty and disaster vulnerability (Suryahadi et al., 2003), security (CSIS, 2016), loss and damages (Warner et al., 2012), impacts on key sectors such as fisheries (USAID Indonesia, 2015), coastal communities (Marfai et al., 2008; Marfai, 2014), food security (Measey, 2012; WFP, 2015) and health (Haryanto, 2009; Ady Wirawan, 2010). Strategies and actions for integrating DRR and CCA needed to be explored further (Djalante et al., 2012), while governance for DRR especially at the local government level has just been initially investigated (Kusumasari et al., 2012).

The next recommendation is on the need to strengthen the capacity of research collaborations between Indonesian and international researchers, multi-disciplinarity of research and publications for high impacts journals. It is clear that some of the very limited Indonesian researchers from ITB, LIPI, and UGM, all in Java, have been involved in international collaborations, and publications of high impacts journal. There are only nine universities, all in Java island, in Indonesia that are within the list of QS World University Rankings, with University of Indonesia tops the list (QS, 2016). Almost all universities in Indonesia have a division within the rectorate and secretariat

that deal with international collaborations. Moreover, the roles of universities and researchers from outside Java had been very limited in their progress. Other universities in the islands of Sumatra, Sulawesi, and Kalimantan and other locations need to put disaster issues as part of their research agendas.

There is a need for better target of scholars to do more collaboration for research and writing for high impact journals. This goes along with strengthening capacity of researchers and lecturers at the universities to write and publish for international journals. The ministry of Education has indeed conduct the scheme of training and giving incentives for lecturers that have published internationally (RISTEKDIKTI, 2016), however, an overall quality and quantity of papers by Indonesian researchers are still much less that those comparable universities in Malaysia or Singapore (RISTEKDIKTI, 2016). There is abundance of materials within Indonesian repositories related to bencana (disaster in English), especially within the repositories with ITB, UGM, and UNSYIAH. These materials and research activities done within the universities needed to be reviewed and submitted for international journals in order to give a broader view on issues that have been discussed by scholars in Indonesia. The Indonesian Association of Disaster Experts was formed in 2014 and has meet annually to discuss their future research guidelines (IABI, 2016). One thing that should be in the agenda is to review current publications in Bahasa Indonesia and collaborations undertaken by Indonesian experts. This will enable better identification of research progress and hence research needs in the future. The list from SCOPUS shows that there is still small numbers of female and of early career researchers.(SCOPUS, 2016b) The first stage is to have proper identification of researchers and make this available to public. The author cannot find repository of researchers from the ministry of education website, let alone determining their progress, history of schooling and research systematically. There have been some concerns to strengthen the capacity of female researchers globally (Larivière et al., 2013), and also similarly in Indonesia. Early career researcher is defined as those who are within 8 years after PhDs or within 6 years of trainings (AHRC, 2016). While globally there has been some systematic efforts to strengthen the capacity of ECR such as trough mentoring (Kram et al., 1985; Clarke, 2004), , there is no clear strategies for the Indonesian ERC done by the Indonesian governments. International journals (Elsevier, 2016) and international and other national research council's (RCUK, 2016) in have allocated resources and funding research specific for ECR.

There is increasing call for a more inter-disciplinarily collaborations so that complex problems on the social and environmental issues can be understood better and problems identifications can target those in needs better (Future Earth, 2016). Although we can see from the list that some of the most prominent authors are not only from universities but also from national level government agencies. The roles of private business and the communities at risk have rarely been part of the research and collaborations. It is also not clear how collaborations amongst scientists from social and physical scientist have taken place in Indonesia. It is also not clear how or whether science (Wagner et al., 2005), policy and industry (Lee, 1996) collaborations have taken place and be documented in these listed publications. These collaborations are important to face complexities of future problems (Leydesdorff et al., 2008), and also to help achieve the outcomes of the Sustainable Development Goals (Nations, 2016) In conclusion this study has been able to determine the progress in research related to hazards, risks, and risk deduction and climate change in Indonesia. It has also been able to examine the roles of Indonesian scientist in collaborations and towards high quality publications. The recommendations are outlined toward these two issues and it is the responsibility both by the Indonesian and international organizations that have and going to work in Indonesia to be able to meet the needs in order for Indonesia to better understood and manage its hazards and risks in the future.

•The decision to develop the two objectives separately is good. However the many sub-chapters created made the paper redundant (in term of results and charts presented) and difficult to read. Reply: The author has rearranged the structure of the paper. Those that are repetitive and cut off and hence a more concise structure is given. The numbers of words is cut from 17,000 to 11, 000. In the original submission,

[Figure]

Section 3 has 2 subsections which are then divided assessments of overall progress and progress based on three research topics. This hence led to lots of repetitiveness in findings. The revised submissions still have 2 subsections, but with much simplified analysis. The main change was that the author removed the sections 3.2.3.2 and 3.2.3.3 and 3.2.3.4 on analysis of research quality on the three major topics since that are all very repetitive.

•This is valid for both the objectives. Reply: The author has rearranged the structure of the paper. Those that are repetitive and cut off and hence a more concise structure is given. The numbers of words is cut from 17,000 to 11, 000.

•A general rearranging of the structure of the paper is needed. Reply: The author has rearranged the structure of the paper. Those that are repetitive and cut off and hence a more concise structure is given. The numbers of words is cut from 17,000 to 11, 000.

•A review paper is a useful tool to give other researchers the state of the art of the current research and advances. It is not just a mere list of the topics of the papers found. Reply: The author notes this comment and tries to add substantial discussion on the drives for those progresses of research.

•As it is impossible to mention all the papers (≈750 are too many) the author needs to justify the methodology of citation (the most recent, the most important, the most cited etc) and provide added comments. Reply: New sentences are added (line 148-152) Data from Scopus are analyzed in terms of time, citation, keywords, and authorships. SCOPUS has within its features the capability for search, discovery and analysis (SCOPUS, 2016a). In this paper, the author uses these features to analyze search results, article metric module, citation overview, and author profile page (SCOPUS, 2016a).

•Regarding the second objective (i.e. authorship) there are too many abbreviations that need to be expressed at least once and an additional explanation is needed for the

provided tables. Reply: The author has ensured that abbreviations are used minimal.

•Moreover, at line 409 the author considered the gender of the authorship of the selected papers. I think this should need more emphasis, consideration and background. Reply: New sentences are added, line 680-692 The first stage is to have proper identification of researchers and make this available to public. The author cannot find repository of researchers from the ministry of education website, let alone determining their progress, history of schooling and research systematically. There have been some concerns to strengthen the capacity of female researchers globally (Larivière, Ni et al., 2013), and also similarly in Indonesia. Early career researcher is defined as those who are within 8 years after PhDs or within 6 years of trainings (AHRC, 2016). While globally there has been some systematic efforts to strengthen the capacity of ECR such as trough mentoring (Kram and Isabella, 1985; Clarke, 2004), , there is no clear strategies for the Indonesian ERC done by the Indonesian governments. International journals (Elsevier, 2016) and international and other national research council's (RCUK, 2016) in have allocated resources and funding research specific for ECR.

5.There is a general lack of discussion in the results section that determines the poor conclusions and recommendations for further research. There is just a mention of the tips for further research that need to be enriched. Reply: The author has completely rewritten the conclusion section

6.Appendix 1 gives no added value to the paper. Reply: Appendix 1 is deleted

7.Figures and tables: • (a) There are too many tables and figures that do not give any additional value to the review. Most of them can be easily replaced with one or two sentences in the text. Reply: The author has reduced the number of tables from 13 to 8, and figures from 26 to 7. All those deleted are discussed in text.

•(b) In many of the bar charts the sum the author provided in the caption does not match the real sum showed by the bars. This bias has been found in some figures and tables. Is this a lack of attention or a justifiable bias? Reply: The author has checked

all captions to make sure that they are accurate.

•́In addition, in Table 5, the citation average of the first row should be 8.21 not 8.0. Please check all of the figures, tables and captions. Reply: The author has checked the table and it is now revised to 8.22 (line 556)

•(c) Generally the captions lacks totally of details, are poor in content and sometimes of agreement. Reply: The author has checked all captions to make sure that they are accurate.

•There are no references of the timeframe, places etc, and some of the charts lack of x or y labels. Reply: The author has added more references to timefame, and places Figure 1, line 16, to add sentence 'Figure 1 Risks map of Indonesia (OCHA-ROAP 2011) showing the Island of Java and Sumatra as most at risks''

Line 54 First is on progress of research on hazards, risks, disasters and climate change in Indonesia, in terms of content and quality, within the timeframe from 1900 to 2016.

Line 75 The second research question is related to the roles of Indonesian authors in contributing for research, international publications and collaborations, within the timeframe from 1900 to 2016.

•(d) The hazard map (Fig.1) presented in the introduction lacks of a legend expressing the colors (supposed to show the level of hazard) and the reference is missing in the reference list. I think that a risk map would be suitable to show the risk profile of the country since Risk is defined by Hazard x Vulnerability x Exposure.

Reply: It is checked again and this is a risk map of Indonesia. A sentence is added (line 16) Figure 1 shows map of risks from natural hazards in Indonesia, showing the islands of Sumatera and Java are most at risks.